# A NEW FRAMEWORK FOR EVALUATING MODEL OUT-OF-DISTRIBUTION GENERALISATION FOR THE BIOCHEMICAL DOMAIN

**Raúl Fernández-Díaz**
School of Medicine
Conway Institute of Biomolecular and Biomedical Research
University College Dublin
Dublin, D04 V1W8, Ireland.

IBM Research
Dublin, D02 PN40, Ireland.
raul.fernandezdiaz@ucdconnect.ie

**Thanh Lam Hoang & Vanessa Lopez**
IBM Research
Dublin, D02 PN40, Ireland.
{t.l.hoang,vanlopez}@ie.ibm.com

**Denis C. Shields**
School of Medicine
Conway Institute of Biomolecular and Biomedical Research
University College Dublin
Dublin, D04 V1W8, Ireland.
denis.shields@ucd.ie

## ABSTRACT

Quantifying model generalization to out-of-distribution data has been a longstanding challenge in machine learning. Addressing this issue is crucial for leveraging machine learning in scientific discovery, where models must generalize to new molecules or materials. Current methods typically split data into train and test sets using various criteria — temporal, sequence identity, scaffold, or random cross-validation — before evaluating model performance. However, with so many splitting criteria available, existing approaches offer limited guidance on selecting the most appropriate one, and they do not provide mechanisms for incorporating prior knowledge about the target deployment distribution(s).

To tackle this problem, we have developed a novel metric, AU-GOOD, which quantifies expected model performance under conditions of increasing dissimilarity between train and test sets, while also accounting for prior knowledge about the target deployment distribution(s), when available. This metric is broadly applicable to biochemical entities, including proteins, small molecules, nucleic acids, or cells; as long as a relevant similarity function is defined for them. Recognizing the wide range of similarity functions used in biochemistry, we propose criteria to guide the selection of the most appropriate metric for partitioning. We also introduce a new partitioning algorithm that generates more challenging test sets, and we propose statistical methods for comparing models based on AU-GOOD.

Finally, we demonstrate the insights that can be gained from this framework by applying it to two different use cases: developing predictors for pharmaceutical properties of small molecules, and using protein language models as embeddings to build biophysical property predictors.

# 1 INTRODUCTION

The last decade has been characterised by the impact that the introduction of machine learning models has had in the acceleration of scientific discovery. These models are frequently used to predict the properties of entities (drug candidates, materials, cells, etc.) that are inherently different from those present in their training distribution. This deployment scenario, known as out-of-distribution (OOD), is particularly frequent within the biochemical domain which encompasses both biological and chemical modelling. Proper OOD evaluation of models is necessary, within this domain specifically, due to the enormous economic and societal impact that wrong predictions might have, for example, when a drug candidate that was predicted as non-toxic (Huang et al., 2021a) goes through to the latter phases of the drug development pipeline, including pre-clinical or clinical trials. Further tasks where OOD evaluation is critical are: modelling of the interaction between a known molecular target and new compounds (Corso et al., 2024), the prediction of the structure of molecular targets mediating disease (Mirdita et al., 2022), or cell type annotation (Fischer et al., 2024). Overall, robust OOD evaluation is necessary for the development of trustworthy predictive models, driving advancements in biochemistry.

Prior literature in the biochemistry domain, has already highlighted the importance of evaluating model generalisation to OOD data, though there is no prior work attempting to build a framework for both measuring the generalisation capabilities of any given model and to provide a statistical method to compare the performance of different models. Instead, prior works have focused on developing algorithms for: i) measuring pairwise similarity values between biochemical entities (Steinegger & Söding, 2017; Rogers & Hahn, 2010; Orsi & Reymond, 2024), ii) splitting a dataset into independent train-test subsets (Butina, 1999; Ramsundar et al., 2019; Tom et al., 2023; Steshin, 2023; Teufel et al., 2023), iii) combining different partitioning algorithms to simulate the composition of a single target deployment distribution (Tossou et al., 2024), and iv) exploring the effect of overlap between training and testing on model performance (Ektefaie et al., 2024). Due to the absence of a comprehensive framework for evaluating model generalisation to out-of-distribution (OOD) data, validating claims about machine learning generalisation remains challenging.

In response to this gap, we first present a framework to study and quantify model generalisation to OOD data for biochemistry. We propose a novel partitioning algorithm that, unlike existing methods (Butina, 1999; Ramsundar et al., 2019; Tom et al., 2023; Steshin, 2023; Teufel et al., 2023), is broadly applicable across various biochemistry contexts, including proteins and small molecules. Our approach is agnostic to the underlying data types; instead, it relies on any similarity functions that can be defined between two data instances. In contrast to previous methods that choose a certain similarity function assuming that it can capture generalisation within the context of a certain modelling task (Butina, 1999; Ramsundar et al., 2019; Tom et al., 2023; Steshin, 2023; Teufel et al., 2023; Tossou et al., 2024; Ektefaie et al., 2024), our framework provides quantitative measurements indicating how appropriate a similarity metric is for guiding the partitioning of a certain dataset, with minimal relience on domain knowledge. We also propose two new similarity functions based on physico-chemical descriptors, rather than structure-based fingerprints and a statistical metric to compare the generalisation capabilities of different models.

# 2 HESTIA-GOOD FRAMEWORK

This section presents the Hestia-GOOD framework for quantifying out-of-distribution generalisation in biochemistry. It is divided into three subsections.

1. We present the AU-GOOD metric for quantifying model generalisation to target deployment distribution(s). This metric corresponds with the expected empirical risk of a model trained on a given dataset with varying train/test similarity conditioned on any arbitrary target deployment distribution(s).

2. We propose an algorithm for dataset partitioning that i) does not remove any data points, ii) creates strict boundaries between training and testing, and iii) supports multiple similarity thresholds.

3. We propose a statistical test to compare the generalisation capabilities of two different models.

## 2.1 AU-GOOD METRIC FOR QUANTIFYING MODEL GENERALISATION TO THE TARGET DEPLOYMENT DISTRIBUTION(S)

We consider the problem of estimating model performance against any arbitrary target deployment distribution(s) that are out-of-distribution of the data available during model training. Our approach consists on three steps:

1. Partition the available data (training data) into training/testing subsets with the condition that there are no elements in the testing set that are similar (up to a threshold value) to those in training. Multiple partitions are created by applying different thresholds to this similarity.

2. The model is trained in each training subset and its performance evaluated against the corresponding testing subset. Then, we describe model performance as a function of the similarity threshold. We refer to this function as the Generalisation Out-Of-Distribution (GOOD) curve.

3. Given any number of arbitrary target deployment distribution(s), we calculate the similarity between each of its elements and the training data. For each element in the deployment distribution, we keep the maximum similarity and calculate a histogram of their distribution such that each bin corresponds to one of the thresholds in the GOOD curve. The AU-GOOD, or Area Under the GOOD curve, can be calculated as the weighted average of all points in the GOOD curve where the weight of each point corresponds to the value of the histogram for the corresponding threshold.

Formally, we demonstrate that the AU-GOOD metric corresponds to the expected empirical risk of the model conditioned on the target deployment distribution (see Appendix - A).

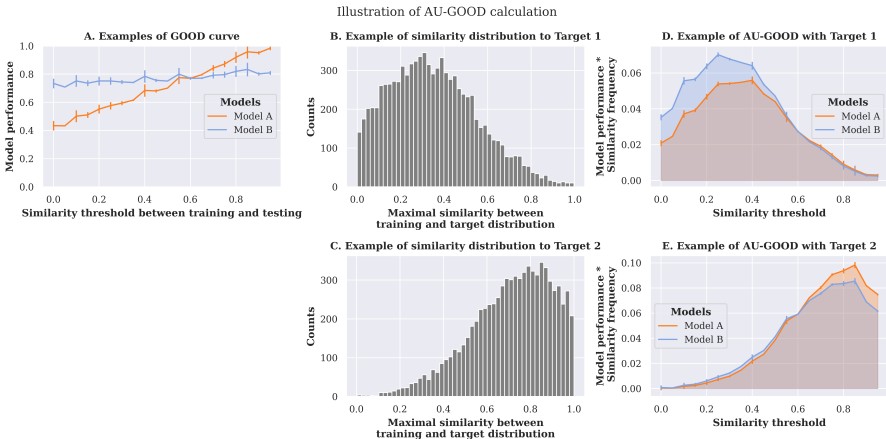

Figure 1: Illustration of the calculation of the AU-GOOD metric. A-C correspond to synthetic data.

To illustrate how the AU-GOOD metric is calculated and what insights can be drawn from its analysis, let us consider a simple simulation, which will serve to illustrate the relationship between GOOD curve, target distribution(s) and AU-GOOD. Let us consider two models, "Model A" and "Model B", with GOOD curves as shown in Figure 1.A. In Figure 1.A, Model A performs better when predicting molecules similar to the training data, whereas for Model B there is comparatively better performance at low similarity values.

Let us now consider two different target deployment distributions, "Target 1" and "Target 2". The histograms reflecting their similarity to the training data are shown in Figures 1.B-C. Target 1 (Figure 1.B) comprises entities with low similarity to the training data, while Target 2 (Figure 1.C) represents the opposite case, where most entities are similar to those the model has been trained on.

The AU-GOOD metric allows for the evaluation of model generalisation relative to a target deployment distribution. Therefore, it allows us to determine that, though apparently Model B has a more consistent performance through different similarity thresholds, Model A is a better choice for Target

2, which is skewed towards molecules similar to the training data. Conversely, Model B is a better choice for Target 1.

This simple example also highlights one of the main strengths of the AU-GOOD metric, which is that it allows to obtain estimators of model generalisation for an arbitrary number of target deployment distributions without the need for additional experiments. This is a significant advantage when compared with other approaches for out-of-distribution performance estimation, like MOOD (Tossou et al., 2024) which focuses on generating test sets that closely resemble the target deployment distribution and requires repeating the training/evaluation cycle for each new target deployment distribution; whereas with ours, only the similarity histogram needs to be updated. Our approach is also similar to Ektefaie et al. (2024) with three main differences: 1) their method defines arbitrarily a certain similarity threshold and considers model performance against the proportion of similar entities between training and testing; whereas, our approach defines a certain proportion of similar entities (0%) and explores the effect on model performance of the different similarity thresholds; 2) their method assumes that whatever similarity function they have chosen is good at capturing the relevant properties that mediate any given task; however, our method makes no such assumption and we provide a series of analyses to determine that a GOOD curve reasonably describes out-of-distribution generalisation; and 3) their approach does not provide any estimation of model performance conditioned on the target deployment distribution(s), which limits its practical application in real-world scenarios.

Finally, our study is limited by the lack of experimental follow up studies to confirm empirically the advantages of our method. The closest study to ours in this regard are the aforementioned MOOD from (Tossou et al., 2024) and Spectra (Ektefaie et al., 2024), neither of which provided a metric to evaluate the approach on "ground truth" data. Development of scenarios where such a ground truth exists will help inform evaluation of the performance of our method.

## 2.2 DATASET PARTITIONING TO SIMULATE OUT-OF-DISTRIBUTION DATA GIVEN A SIMILARITY VALUE

The AU-GOOD values can also be affected by the sampling performed during the partitioning step, as it may bias the composition of the test set. There are three conditions that the partitioning algorithms have to fullfil are: 1) to not remove any data point, otherwise the comparison between thresholds could also correspond to the loss of information and would make them not directly comparable; 2) to enforce strict boundaries between training and testing subsets for any given similarity threshold (Walsh et al., 2020), otherwise we would not be measuring strictly the dependence on the similarity threshold, but also (implicitly) the effect of overlap within any given threshold; and 3) to allow for multiple thresholds, otherwise we cannot describe model performance as a function of the threshold.

All prior partitioning algorithms described for biochemical data violate at least one of the three conditions: perimeter and maximum dissimilarity do not fulfill condition 2 and scaffold split does not fulfill condition 3 (Ramsundar et al., 2019), cluster splits relying on hierarchical clustering (Tom et al., 2023) do not fulfill conditions 2 and 3; GraphPart (Teufel et al., 2023) or Lo-Hi splitting (Steshin, 2023) do not fulfill condition 1. The Butina split (Butina, 1999) generates clusters where every member of a given cluster is guaranteed to be closer to the centroid than the threshold, however the frontiers between clusters are not strict, and therefore does not fulfill condition 2.

Here, we propose the CCPart (**C**onnected **C**omponents **Part**itioning) a new strategy for partitioning the datasets that fulfills all three previous conditions and that is extensible to any biochemical data. The algorithm first constructs a graph, where the nodes are all elements in the datasets and the edges are drawn between nodes with similarity greater than the threshold. It then identifies all unconnected subgraphs. The testing set is iteratively built by assigning the smallest subgraphs until the desired size is reached. Smaller subgraphs are considered more unique. CCPart thus focuses the evaluation on the most dissimilar regions of the dataset distribution. Additionally, stratified sampling can be applied to maintain label balance in classification tasks. Figure 2 illustrates this process (see Algorithm S1).

In any case, our framework is agnostic to the specific partitioning strategy used and future development of alternative algorithms that fulfill the three conditions could improve the accuracy of the

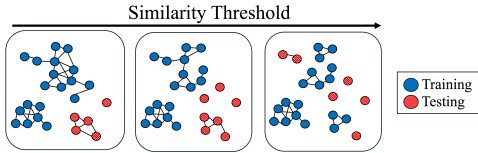

Figure 2: Schema showing how CCPart builds the training and testing subsets based on the similarity graph at different similarity thresholds.

AU-GOOD calculation. It is also an open research question what the impact is of violating any of the three assumptions on the reliability of the AU-GOOD metric.

## 2.3 STATISTICAL TEST FOR COMPARING AU-GOOD VALUES

Generally, comparison between the statistical significance of the difference in performance between two given models is made by comparing their confidence intervals. This standard practice, implicitly assumes that performance is normally distributed. However, the performance values at different thresholds cannot be assumed to be normally distributed. Therefore, the analysis of the statistical significance between individual AU-GOOD values requires a different kind of statistical test. We propose to use the non-parametric equivalent to the one-tailed T-test, the one-tailed paired Wilcoxon ranked-sum test Wilcoxon (1945); Ash et al. (2024). Finally, if we want to compare the performance of $n$ models, $n \times n$ pairwise tests can be performed. In cases where we are performing multiple tests ($n > 5$), the Bonferroni correction for multiple testing (Bonferroni, 1936) can be applied so that the significance threshold depends on the number of models: $0.05/n$ (Ash et al., 2024).

## 2.4 HESTIA: COMPUTATIONAL EMBODIMENT

The partitioning algorithms rely on pairwise similarity calculations and scale in complexity and memory footprint by $\mathcal{O}(N^2)$. In response to this, our implementation improves computational and memory efficiency by relying on sparse matrices and parallelism. We have run toy examples with datasets of up to 400,000 protein sequences within a day in 32 CPU cores and 200GB of RAM, thus we believe that scaling does not limit the usefulness of the method within the domain. In other domains, algorithmic improvements might be required for scaling to very large datasets.

We have open sourced the code [1], including: 1) wrappers for most common similarity functions between a) small molecules, b) biosequences including protein and nucleic acids, c) protein structures, and d) embeddings from pre-trained representation learning models; 2) implementation of different partitioning algorithms such as: a) GraphPart and b) CCPart; and 3) calculation of AU-GOOD values. More details in Appendix - B.

## 3 EXPERIMENTS

We have chosen two different real-world use cases to demonstrate the insights that can be drawn from using our framework for the comparison of ML models within the biochemical domain. The first one is the selection of predictive models for pharmaceutical properties given a target deployment distribution and the second is the selection of what protein language model embedding to use for different target deployment distributions.

## 3.1 USE CASE 1: PREDICTION OF PHARMACEUTICAL PROPERTIES OF SMALL MOLECULES

ADMET (Absorption, Distribution, Metabolism, Excretion, and Toxicity) properties encompass different biophysical attributes of small molecules that may affect both their pharmacokinetic and pharmacodynamic behaviour within the organism (Wang et al., 2024). The estimation of which model will generalise better to which target deployment distribution, will allow researchers to decide on what model to use for each particular project. For the purposes of this study, we have selected the

---

[1]Github Repository: https://github.com/IBM/Hestia-GOOD

Drug Repurposing Hub (Corsello et al., 2017) as the library of choice as it represents a realistic scenario with known approved drugs.

We examine our framework with six datasets containing different ADMET properties from the Therapeutics Data Commons collection (Huang et al., 2021b): Ames' mutagenicity (Xu et al., 2012), cell effective permeability in Caco-2 cell line (Wang et al., 2016), drug-induced liver injury (DILI) (Xu et al., 2015), acute toxicity LD50 (Zhu et al., 2009), drug half-life (Obach et al., 2008), and parallel artificial membrane permeability assay (PAMPA) (Siramshetty et al., 2021). We report model performance with Matthew's correlation coefficient for classification tasks (Ames, DILI, and PAMPA) and Spearman's $\rho$ correlation coefficient for regression tasks (Caco2, Half-life, and LD50).

Each experiment described uses as featurization method Extended-Connectivity FingerPrints (ECFP) with radius 2 and 2,048 bits, as they are a well-known baseline for all the datasets under consideration (Huang et al., 2021b). Each experiment consists of 5 independent runs with different seeds. Each run for a dataset and similarity metric has the exact same partitions, because the partitioning algorithm is heuristic and not stochastic, therefore any variance between runs depends on the learning algorithm and the Bayesian hyperparameter-optimisation which is conducted for each run independently with Optuna (Akiba et al., 2019) (see Table - S3).

### 3.1.1 What is the best similarity function?

**Experimental setup**

The first step for using the Hestia-GOOD framework is to determine what is the optimal similarity function to use. We evaluated three types of fingerprints: ECFP with radius 2 and 2,048 bits, MACCS, and chiral Min-hashed atom pair (MAPc) with radius 2 and 2,048 bits. We considered 4 binary set similarity functions for the binary fingerprints (ECFP and MACCS): Tanimoto (Rogers & Tanimoto, 1960), Dice (Dice, 1945), Sokal (Sokal & Michener, 1958), and Rogot-Goldberg (Rogot & Goldberg, 1966). We also considered cosine similarity as an alternative geometry-based similarity function. For MAPc, which is not a binary fingerprint, we only considered the Jaccard similarity. Additionally, we have also considered alternative similarity metrics that are not fingerprint-based, including: the euclidean distance between embeddings from two pre-trained chemical language models: Molformer (Ross et al., 2022) and ChemBERTa-2 (77M MLM) (Ahmad et al., 2022); the canberra distance between two types of vectors of physicochemical descriptors 1) BUTCD (Stanton, 1999) and Lipinski (Lipinski et al., 1997). More details in Appendix - C.

**Criteria for selecting similarity function**

We observed that different similarity functions led to GOOD curves with notably different shapes (see Appendix - D). The two properties of the curves that experienced the biggest variance across similarity functions were 1) their dynamic range, i.e., the difference between the minimum and maximum similarity thresholds that could be used to generate viable partitions with test sets with at least 18.5% of the total data[2]; and 2) their monotonicity, measured as the Spearman's $\rho$ correlation coefficient between similarity threshold and model performance. A necessary assumption of the GOOD and AU-GOOD calculations is that model performance can be described as a function of the similarity threshold. Therefore, Spearman's correlation coefficient serves as a robust, non-parametric diagnostic test to ensure that any GOOD curve meets this assumption, without assuming linearity.

We propose to use both metrics as selection criteria:

1. Rank all similarity functions by their largest dynamic range and choose the top functions.

2. Compare the monotonicity of the GOOD curves generated with those similarity functions and select the similarity function with the highest monotonicity.

The first criterion aims to produce GOOD curves that span a greater similarity range to minimise the impact of noise. The second focuses on improving discrimination between low and high similarity thresholds for better evaluation of model generalization. It also serves as a diagnostic test to ensure that the GOOD curves meet the assumption of model performance dependence on the similarity thresholds. If none of the similarity functions achieves a reasonable monotonicity, it is likely that a

---

[2]We keep the conventional 20% value and allowed for a 1.5% margin of error.

custom similarity function is required; or that there are underlying issues with the data generation process that need to be corrected in a dataset specific way.

**Choice of similarity function is dataset-specific**

We applied these criteria to the datasets and similarity metrics considered in the study. Table 1 summarises the decisions and the properties of the GOOD curves obtained in each case and Appendix - D contains a more in-depth analysis.

Overall, MAPc - Jaccard seems to be the most versatile similarity metric, being the optimal choice for three out of six datasets. It has the highest dynamic range and good monotonicity in the majority of datasets. This is consistent with the literature that describes it as a fine-grained and sensitive tool for performing similarity searches in chemical databases (Orsi & Reymond, 2024).

It is also interesting that Molformer and ChemBERTa-2, both similarity metrics based on the homonymn molecular language models, show the best relative performance in the same two datasets PAMPA and Caco2. This suggests that the models are able to capture chemical properties relevant for the respective tasks that are then exploited when partitioning. It also worth noting that, in both cases, the Molformer similarity has larger dynamic range, demonstrating a greater resolution at low thresholds.

Finally, one of the new similarity metrics proposed in this study, Lipinski, is the best for the half-life dataset which proved quite challenging for the other alternatives.

Table 1: Similarity function chosen per dataset.

| Dataset | Similarity metric | Dynamic range | Monotonicity |
|---|---|---|---|
| Ames | MAPc - Jaccard | 0.75 | $0.82 \pm 0.02$ |
| DILI | MAPc - Jaccard | 0.85 | $0.81 \pm 0.02$ |
| PAMPA-NCATS | Molformer | 0.80 | $0.13 \pm 0.02$ |
| Caco2 | Molformer | 0.85 | $0.76 \pm 0.03$ |
| Half-life | Lipinski | 0.60 | $0.46 \pm 0.06$ |
| LD50 | MAPc - Jaccard | 0.70 | $0.94 \pm 0.02$ |

The main conclusion that can be drawn from these results is that there is no one-size-fits-all similarity function. Different datasets require different similarity metrics. Therefore, before starting any analysis regarding model generalisation, a careful selection of the similarity metric used to measure generalisation is crucial. This initial step of analysis is missing from previous works (Butina, 1999; Ramsundar et al., 2019; Steshin, 2023; Tom et al., 2023; Tossou et al., 2024; Ektefaie et al., 2024).

Remarkably, the PAMPA dataset has low monotonicity with all similarity metrics considered in this study, which makes the interpretation of the GOOD curve unreliable. We believe that this behaviour is due to the idiosyncrasies of this dataset which is relatively small ($\approx 2,000$ molecules) and highly unbalanced (1,739 positives to 295 negatives). The imbalance is further exacerbated by the partitioning algorithm with the test set containing, in most partitions, around two thirds of all negatives $\approx 200$. This dataset provides a concrete showcase of the limitations of our framework, which are: 1) it depends on the similarity metrics and certain tasks may require custom similarity functions and 2) it may not be reliable for highly unbalanced datasets as the partitioning can increase the imbalance. For this reason, the PAMPA dataset will be excluded from the next step, as the GOOD curve and AU-GOOD values are not reliable.

In any case, we highly recommend performing an initial analysis with any similarity function to determine whether the GOOD has an appropriate dynamic range and meets the assumption of correlation between similarity threshold and model performance.

**Qualitative analysis of CCPart as an OOD partitioning algorithm**

We visualised the overlap between training and testing partitions (generated with the corresponding chosen similarity metric). The visualisation of the chemical space is achieved by using uniform manifold approximation and projection (UMAP) (McInnes et al., 2018; Ross et al., 2022; Ahmad et al., 2022) to project the ECFP fingerprints of each dataset into two dimensions. Figure 3 shows how the testing partition of the Cell-effective permeability dataset gets clustered together at low thresholds and gets more evenly distributed over the training data distribution as the threshold increases. It is

particularly interesting how the partitions at threshold 0.9 are still less evenly distributed than those obtained with random partitioning. Figure S2 contains similar representations for the rest of the datasets considered in the study.

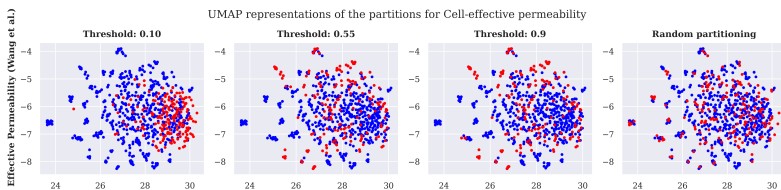

Figure 3: UMAP representation of the chemical space covered by the training (blue) and testing (red) partitions of the Cell-effective permeability dataset at different similarity thresholds.

Overall, these visualizations provide a qualitative and empirical demonstration that the CCPart algorithm is able to generate out-of-distribution partitions.

### 3.1.2    What models generalise better to the Drug Repurposing Hub?

**Experimental setup**

We evaluated the performance of four traditional ML algorithms to build models for the ADMET datasets: K-nearest neighbours (KNN), support vector machines (SVM), random forest (RF), and light gradient boosting (LightGBM) and evaluated their AU-GOOD values using as target deployment distribution the Drug Reporpusing Hub Corsello et al. (2017).

**Demonstration of an in-depth Hestia analysis**

To demonstrate the depth of the analysis that can be performed with our framework, let us consider one of the datasets in detail. Figure S3 contains equivalent analyses for the rest of the datasets. Figure 4.A shows the GOOD curves for all models considered. The behaviour of KNN is particularly interesting as it tends to perform worse at lower similarity thresholds and better at higher similarity. This observation is not surprising in itself (KNN explictly depends on the distance to neighbouring data points within the data), but it exemplifies the type of situation our framework is built to address.

Figure 4.B shows the distribution of maximal similarities between training and target deployment distribution. In this particular case, it seems that the distribution is skewed towards higher similarities. Figure 4.C shows the distribution of AU-GOOD values of the Matthew's correlation coefficient calculated for each model against the target deployment distribution. It shows that RF has the best average AU-GOOD value, even though it shows greater variance across runs. Interestingly, KNN is the second best model, despite it performing worse at the lower similarity thresholds. This is because the maximal similarity distribution is skewed towards higher similarities and, therefore, the performance at those thresholds has a greater contribution towards the final AU-GOOD score.

Figure 4.D shows the p-values of comparing the 4 models against each other with the Wilcoxon signed-rank test. In this case, the alternative hypothesis is that Model A (y-axis) has better AU-GOOD values than Model B (x-axis). This clearly shows that RF is significantly better ($p < 0.001$) than the other three models.

**Demonstration of a summary of the Hestia analysis**

The results for the rest of the datasets are summarised in Table 2. The significant rank reflects the Wilcoxon signed-rank test results. For each model, it is calculated as the difference between the total number of models and the number of models to which it is significantly superior. For example, the significant rank for RF in Figure 4 would be 4 (number of models) - 3 (number of models that are significantly inferior to RF) = 1; KNN would be $4 - 2 = 2$, SVM and LightGBM would be $4 - 0 = 4$.

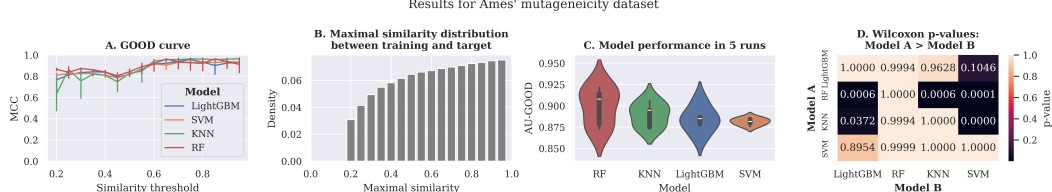

Figure 4: HESTIA analysis for Ames' mutageneicity dataset. A. GOOD curve with MAPc (radius 2 and 2,048 bits) and jaccard index; B. Maximal similarity between entities in Ames' mutageneicity dataset and target distribution (Drug Repurposing Hub); AU-GOOD values across 5 different runs; Wilcoxon test p-values with alternative hypothesis Model A > Model B.

Table 2: Comparison between the different downstream models for each dataset. AU-GOOD refers to average AU-GOOD value across 5 runs, the error corresponds to the standard error of the mean. The AU-GOOD value is calculated for the classification tasks for the Matthew's correlation coefficient (MCC) for Ames, DILI, and PAMPA. For the regression tasks ( Caco2, Half life, and LD50) Spearman's $\rho$ is used. S. Rank: Significant rank.

| | KNN | | SVM | | RF | | LightGBM | |
|---|---|---|---|---|---|---|---|---|
| Dataset | AU-GOOD | S. Rank | AU-GOOD | S. Rank | AU-GOOD | S. Rank | AU-GOOD | S. Rank |
| Ames | $0.89 \pm 0.01$ | 2 | $0.88 \pm 0.00$ | 4 | $0.90 \pm 0.01$ | 1 | $0.88 \pm 0.01$ | 4 |
| DILI | $0.95 \pm 0.00$ | 3 | $0.96 \pm 0.00$ | 1 | $0.95 \pm 0.00$ | 3 | $0.77 \pm 0.04$ | 4 |
| Caco2 | $0.87 \pm 0.00$ | 4 | $0.92 \pm 0.00$ | 1 | $0.92 \pm 0.00$ | 2 | $0.88 \pm 0.00$ | 3 |
| Half life | $0.93 \pm 0.00$ | 2 | $0.97 \pm 0.00$ | 1 | $0.69 \pm 0.04$ | 3 | $0.23 \pm 0.01$ | 4 |
| LD50 | $0.86 \pm 0.00$ | 2 | $0.83 \pm 0.00$ | 4 | $0.89 \pm 0.00$ | 1 | $0.83 \pm 0.00$ | 4 |
| Average | $0.90 \pm 0.01$ | 2.6 | $0.91 \pm 0.01$ | 2.2 | $0.87 \pm 0.02$ | **2.0** | $0.72 \pm 0.05$ | 4.8 |

## 3.2 USE CASE 2: EVALUATION OF PROTEIN LANGUAGE MODELS

Protein Language Models (PLMs) are pre-trained representation learning models that rely on the transformer architecture and are trained with a masked-language modelling objective on protein sequences. Here, we explore which PLM generates an embedding space that is better suited for generalising to different target deployment distributions. We considered a biophysical property prediction task, Optimal temperature for catalysis (Li et al., 2022), from (Chen et al., 2024). To this end, we downloaded all protein sequences for four organisms in the SwissProt database (Consortium, 2019) as target deployment distributions: human (*Homo sapiens*), Baker's yeast (*Saccharomyces cerevisiae* strain YJM789), and bacteria (*Escherichia coli* strain CFT073 / ATCC 700928 / UPEC) and the human hepatitis B virus (HPV-B) *Orthohepadnavirus hepatitis B virus*.

### 3.2.1 EXPERIMENTAL SETUP

We considered the following models: ESM2-8M (Lin et al., 2022), ESM2-150M (Lin et al., 2022), ESM2-650M (Lin et al., 2022), ProtBERT (Elnaggar et al., 2021), Prot-T5-XL (Elnaggar et al., 2021), Prost-T5 (Elnaggar et al., 2021), Ankh-base (Elnaggar et al., 2023), and Ankh-large (Elnaggar et al., 2023). We extracted the embeddings for all sequences in our downstream datasets and for each sequence $s$ we computed its embeddings using average pooling (Unsal et al., 2022). We used SVM as the downstream model to reduce the amount of noise in our analyses due to model selection.

### 3.2.2 WHAT IS THE BEST SIMILARITY FUNCTION?

The task we are considering concerns protein sequence property prediction. We decided to compare two computationally efficient similarity functions MMSeqs2 and MMSeqs2 with prior k-mer prefiltering (Steinegger & Söding, 2017). However, for other protein tasks where data points are evolutionary related we recommend using alternative metrics like Hamming distance like in the Fluorescence task in the TAPE benchmarks Rao et al. (2019) or distance in phylogenetic trees.

The results clearly indicate that MMSeqs2 with prefilter is the best option in all respects i) dynamic range, (Figure S4.A) and ii) monotonicity (Figure S4.B).

### 3.2.3 What PLM generates the most useful embeddings?

Figure 5 shows the analyses of the different PLM embeddings (More details in Figure S5). Clearly, the bigger models (Prot-T5-XL, ESM2-650M, and Prost-T5) tend to generalise better than their smaller counterparts. This is particularly clear with the ESM2 models which are ordered in ascending performance according to their size ESM2-8M, ESM2-150M, and ESM2-650M. The worst performing models are the Ankh models, which is surprising as the original paper (Elnaggar et al., 2023) reported better performance in several tasks. This result is not conclusive as we are only evaluating a single task. However, it is an interesting finding that opens the question as to whether scaling down model parameter size might be detrimental for the ability of the model embeddings to generalise to new data.

It is also worth noting that, although the ranking of the models is the same regardless of the target distribution, in some cases, like human or the hepatitis virus, the use of a much smaller model like ESM2-150M instead of Prost-T5 could be statistically justified.

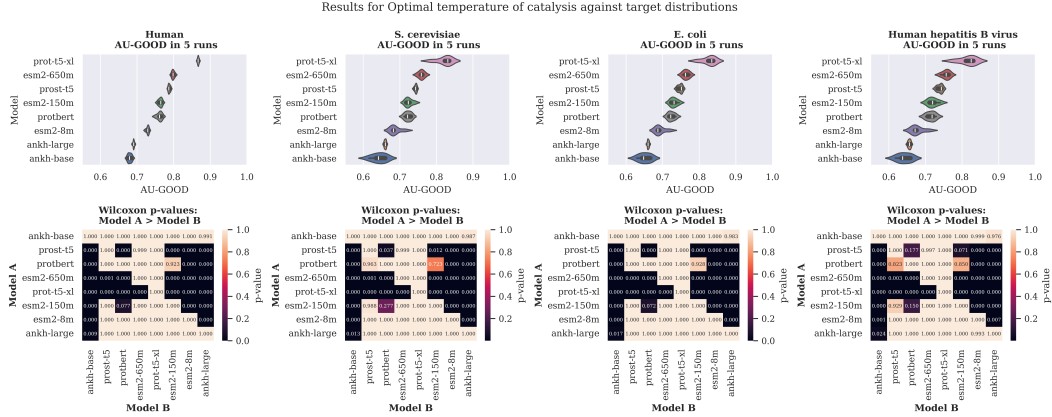

Figure 5: Hestia analyses for Optimal Temperature of Catalysis, comparing different PLM representations. AU-GOOD values correspond to Spearman's $\rho$. The variance corresponds to 5 different runs.

## 4 Conclusion

The AU-GOOD is a new metric that estimates the expected model performance against any target deployment distribution(s), and thus provides a quantifiable value to guide the selection of the most appropriate model to use in different deployment scenarios, without the requirement for additional experiments. It is applicable to any biochemical entities for which a relevant similarity function can be defined. The calculation of this metric requires partitioning the training data into training/testing subsets that are increasingly dissimilar. We propose CCPart a new partitioning algorithm to generate strict splits at multiple thresholds, without the removal of any data points.

We provide a robust framework for analysing whether a similarity function is appropriate for partitioning a given dataset, with a series of quantitative metrics to provide a formal analysis of the reliability of the resulting GOOD curves. Finally, we discuss how to obtain statistical support for comparing different AU-GOOD values. Our experiments show that different datasets may require different similarity functions that better capture the chemical relationships in the context of the specific modelling problem and that the selection of the appropriate similarity metric for each dataset has a significant impact on the GOOD curves.

We demonstrate the use of this framework for two different use cases: the development of models for predicting properties of small molecules, and the selection of a protein language model to generate embeddings upon which to build biophysical property predictors.

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

# A  MATHEMATICAL FORMALISATION OF AU-GOOD METRIC

This Appendix section discusses the formal definition of the AU-GOOD metric and its connection with statistical learning theory. It is divided into four subsections, as follows: i) empirical risk in terms of the dataset partitioning strategy, ii) definition of the AU-GOOD metric as an estimation of model generalisation to a target distribution, iii) theoretical considerations regarding the assumption of unbiased sampling during dataset partitioning, and iv) empirical considerations regarding model performance metrics.

## A.1  EMPIRICAL RISK IN TERMS OF THE DATASET PARTITIONING STRATEGY

This subsection demonstrates that the empirical risk can be expressed in terms of the partitioning operation used to divide the data into training and testing subsets.

Let us consider a model $f_\theta$ defined by a set of parameters $\theta$ and trained on a data distribution $P(x,y)$ where each datum is defined by a set of features $x \in \mathcal{X}$ and a set of labels $y \in \mathcal{Y}$. The model will attempt to approximate an unknown function $g$ that provides a mapping from the feature space to the label space $f_\theta \approx g \colon \mathcal{X} \to \mathcal{Y}$. The quality of the approximation can be described in terms of a loss function $\mathcal{L}$ that measures the error in each individual mapping. In the end, the parameters of the model will be optimised to minimise the expectation of the loss across the whole population which is also known as the population risk:

$$\mathcal{R}(f_\theta) := \mathbb{E}_{(x,y) \sim P(x,y)}[\mathcal{L}(f_\theta(x), y)] \tag{1}$$

In practice, we do not have access to the data distribution, only to a subset, $\mathcal{D} \sim P(x,y)$. Without access to the data distribution, the population risk has to be approximated by the empirical risk. The empirical risk of a model given a dataset $\mathcal{D}$ is calculated by partitioning $\mathcal{D}$ into two mutually exclusive subsets $\mathcal{T}, \mathcal{E} \in \mathcal{D}$ such that $\mathcal{T} \cap \mathcal{E} = \emptyset$, where $\mathcal{T}$ is the train subset and $\mathcal{E}$ is the test subset defined by the partitioning operation denoted as $\Phi \colon \mathcal{D} \to \mathcal{T}, \mathcal{E}$. The model trained on the train subset is denoted as $f_{\theta,\mathcal{T}}(x)$ Therefore, the empirical risk can be estimated in terms of the partitioning strategy:

$$\hat{\mathcal{R}}_\mathcal{D}(f_\theta) := \frac{1}{n} \sum_{(x,y) \in \mathcal{E}} \mathcal{L}(f_{\theta,\mathcal{T}}(x), y) \tag{2}$$

where $n$ is the number of elements in $\mathcal{E}$.

## A.2  DEFINITION OF A METRIC ESTIMATING MODEL GENERALISATION TO A TARGET DISTRIBUTION

This subsection builds upon the previous definition of empirical risk in terms of the partitioning operator and derives a metric for model generalisation to any target data distribution based on the similarity between the elements of the training and target distributions.

Let us consider a hypothetical function that measures the similarity between the features of any two elements in the population, $s \colon \mathcal{X} \times \mathcal{X} \to \mathbb{R}$. Let us also consider a partitioning operator that enforces a maximal value of similarity $\lambda$ between the elements in training and testing. Such an operator would necessarily be a function of both the similarity function $s$ and the threshold similarity $\lambda_s$. Then, the empirical risk would depend on the similarity function and the similarity threshold:

$$\hat{R}_{\lambda_s}(f_\theta) := \frac{1}{n} \sum_{i \in \mathcal{E}} \mathcal{L}(f_{\theta,\mathcal{T}}(x_i), y_i) \tag{3}$$

Model generalisation to a target deployment distribution $\mathcal{E}^*$ that has been drawn from a population distribution $\mathcal{E}^* \sim P^*(x,y)$ different from $P(x,y)$ could then be described as the expectation of the empirical risk across the distribution of the similarity $\lambda_s$, denoted as $P(\lambda_s | P^*)$ induced from the unknown target distribution $P^*(x,y)$, within the bounds of a minimal similarity $\lambda_0$ and a maximal similarity $\lambda_n$:

$$\mathcal{G}_\Phi(f_\theta|P^*) \coloneqq \mathbb{E}_{[\lambda_s|P^*]}\hat{R}_{\lambda_s}(f_\theta) = \int_{\lambda_0}^{\lambda_n} \hat{R}_{\lambda_s} P(\lambda_s|P^*)d\lambda_s \qquad (4)$$

where $P(\lambda_s|P^*)$ is the probability density distribution of the similarity between $\mathcal{D}$ and $\mathcal{E}^*$, i.e., the expected distribution of similarities between $\mathcal{D}$ and $\mathcal{E}^*$: $P(\lambda_s|P^*) = P(\max\{s(x_i, x_j) \ \forall \ x_i, x_j \in \mathcal{D}, \mathcal{E}^*\} = \lambda_s)$. This integral can be approximated numerically by:

$$\mathcal{G}_\Phi(f_\theta|P^*) \approx \sum_{i=\lambda_0}^{\lambda_n} \hat{R}_{\lambda_s^i} P(\lambda_s^i|P^*)\Delta\lambda_s \qquad (5)$$

## A.3 SAMPLING CONSIDERATIONS

In this subsection, we discuss the limitations of the assumption of unbiased sampling made in the previous subsections.

In many biological domains, data points are noisy, and it's quite possible that the type of disconnected subgraphs generated by similarity-based partitioning algorithms would tend to bias the testing set towards unrepresentative regions of the (bio)chemical space. This behaviour would lead to unstable GOOD curves where there is low monotonic correlation between similarity threshold and model performance and would therefore be identified when following the guidelines outlined in Subsection 3.1.1.

However, if the GOOD curve is monotonic, there could be some specific partitions at given thresholds that show unstable behaviour. Given that the AU-GOOD is an expected value, with a sufficient number of thresholds ($n > 10$) and a wide enough dynamic range, the estimated and the true population AU-GOOD values will tend to converge.

## A.4 MODEL PERFORMANCE METRIC CONSIDERATIONS

Finally, in this subsection we discuss practical limitations, regarding the effect that choice of model performance metric might have on the calculated AU-GOOD values and our general recommendation.

The GOOD curve generates test partitions with different label distributions at different thresholds. Thus, both the monotonicity and AU-GOOD calculation could be affected by the label distribution shifts across different thresholds.

For this reason, we recommend the use of model performance metrics that are not sensitive to label imbalance, like Matthew's correlation coefficient or weighted F1 for classification tasks and Spearman's or Pearson's correlation coefficient for regression tasks.

## B HESTIA COMPUTATIONAL SUITE

### B.1 SIMILARITY CALCULATION

Hestia implements a diverse array of pairwise similarity functions $s(x, y)$ which need to fulfill two conditions:

1. The similarity between an entity and itself is maximal.
2. They are symmetric, $s(x, y) = s(y, x)$.

**Biological sequences.** The similarity function calculated for protein sequences and nucleic acids is the sequence identity in pairwise alignments. Local alignments are calculated using the MMSeqs2 implementation (Steinegger & Söding, 2017) of the Waterman-Smith algorithm (Smith et al., 1981) with or without prior k-mer prefiltering. Global alignments are calculated using the EMBOSS implementation (Rice et al., 2000) of the Needleman-Wunch algorithm (Needleman & Wunsch, 1970). More information regarding the empirical differences between local and global alignments can be

found in (Polyanovsky et al., 2011; Teufel et al., 2023). The denominator used to calculate the identity can be the length of the longest or the shortest sequence, as well as the length of the full alignment. Choice of the most appropriate denominator depends on the dataset (May, 2004).

**Protein structures.** The similarity function calculated for a pair of protein structures is the probability that they belong to the same SCOPe family (Chandonia et al., 2019). This probability is approximated using the Foldseek alignment algorithm (van Kempen et al., 2022) with both sequence and structural interaction representation (3Di+AA) in either global (Foldseek-TM) or local (Foldseek) modes.

For both biological sequences and protein structures, pairwise alignments are not necessarily symmetric so we enforce condition 3 by taking the maximal similarity for each pair of entities: $s^* \coloneqq max[s(x, y), s(y, x)]$.

**Small molecules.** Similarity between small molecules can be calculated using a variety of similarity functions including: Tanimoto (Rogers & Tanimoto, 1960), Dice (Dice, 1945), Rogot-Goldberg Rogot & Goldberg (1966), Sokal (Sokal & Michener, 1958), euclidean, manhattan, canberra, or cosine similarity between various types of fingerprints including extended connectivity fingerprints (ECFP) (Rogers & Hahn, 2010), MACCS (Durant et al., 2002), or MAPc (Orsi & Reymond, 2024); an physicochemical descriptors.

**Representation learning pre-trained model embeddings.** Similarity between embeddings includes traditional geometrical distance functions like cosine, euclidean, manhattan, canberra, as well as offering the option to use any custom similarity function. The distance functions $d$ are transformed into similarities with the following expression: $s(x, y) = \frac{1}{1+d(x,y)}$

### B.2 SIMILARITY CORRECTION ALGORITHMS

**Basic notation.** Let $D$ be an arbitrary dataset comprised of $n$ entities. $D$ can be expressed as a graph $G(N, E)$ where the nodes ($N$) are the set of all entities in $D$ and the edges ($E$), the subset of all pairwise similarity measurements between those entities $s(N_i, N_j)$ with values above a threshold $\lambda$, thus, $E = \{(n_1, n_2) \,\forall\, n_1, n_2 \in N$ such that $s(n_1, n_2) > \lambda\}$. Let $T$ and $V$ be two partitions of $D$ such that $T \cap V = \emptyset$. Then, $E_f$, the forbidden edges, can be defined as the subset of all similarities between any two entities in $T$ and $V$ that is above the threshold, $E_f = \{(t, v) \,\forall\, t \in T, v \in V$ such that $f(t, v) > \lambda\}$.

In this paper, similarity correction techniques refer to algorithms targeting the reduction of $E_f$. We discuss them extensively as outlined below.

**Similarity reduction.** Similarity reduction aims at reducing redundant entities from $D$. This is achieved by a two-step process comprising a clustering step and a redundancy reduction step in which the representative entities of each cluster are selected and the rest of cluster members are removed. Hestia relies on custom implementations of greedy incremental clustering and greedy linear cover-set algorithms for the clustering step. These algorithms are commonly used in the context of sequence clustering, by specialised software like CD-HIT (greedy incremental clustering) (Fu et al., 2012) and MMSeqs (both) (Steinegger & Söding, 2017). Our custom implementations generalise their utility to any arbitrary data type for which a pairwise distance matrix can be calculated.

**Random partitioning.** Random partitioning algorithms aim to divide the dataset into subsets through unbiased sampling. The idea is to generate partitions with similar distributions. Strictly speaking, it could only be considered a similarity correction algorithm under the assumption of independent and identically distributed data, e.g., after performing similarity reduction on the dataset. Hestia leverages the corresponding `scikit-learn` implementation (Kramer, 2016) of the algorithm.

**GraphPart generalisation.** Similarity partitioning algorithms aim at dividing the dataset into $n$ subsets or partitions such that $E_f = \emptyset$ between any two partitions. This is achieved through the removal of entities whenever necessary. Hestia relies on a custom implementation of the Graph-Part algorithm (Teufel et al., 2023) that generalises it to any similarity function and biochemical data type.

The algorithm starts with a clustering step using limited agglomerative clustering with single linkage with the restriction that the clustering stops when either a) a cluster reaches the expected partition

size $N/n$ where $N$ is the number of entities in the dataset and $n$ the number of desired partitions or b) there are no more edges above the threshold $\lambda$. Then, clusters are iteratively merged into the $n$ partitions so that the generated partitions have a balanced distribution of labels (if the dataset has categorical labels) and similar number of entities. The number of interpartition neighbours (entities with $E > \lambda$) is checked and entities are moved to the partition with which they have the most neighbours. If the number of entities with at least one interpartition neighbour is greater than 0, then the $c \times \log{(i/100)} + 1$ entities with most interpartition neighbours are removed, where $c$ is the number of interpartition neighbours and $i$, the current iteration. This iterative process continues until $c = 0$. Our custom implementation leverages `Scipy` (Virtanen et al., 2020) and `Networkx` (Hagberg & Conway, 2020).

**Connected components partitioning.**

The algorithm first identifies the set of all unconnected subgraphs of $G$, $\{U_1, U_2, \ldots, U_k\}$. These unconnected subgraphs, by definition, will not have any intercluster neighbours, otherwise they would not be unconnected. We have observed that generally this strategy leads to one cluster being populated with most entities and a variable number of much smaller subgraphs ($10$ - $10^3$). The smaller a cluster is, the more unique it is with respect to the dataset distribution. Based on this assumption, the algorithm builds the evaluation set by assigning subgraphs in ascending order of number of members, i.e., smaller subgraphs first. As above, there is also the optional additional objective of maximising evaluation label balance. This sampling strategy biases the evaluation subset towards the regions of the dataset distribution that are the most unconnected and thus the most dissimilar to other members of the dataset. See Algorithm - 1.

---

**Algorithm 1** CCPart (Connected Components Partitioning) algorithm

---

1: Define $G(\mathcal{D}, \lambda_s)$
2: U $\leftarrow$ Find all unconnected subgraphs of G
3: Sort U in order of ascending number of elements
4: $\mathcal{T}, \mathcal{E} \leftarrow \emptyset, \emptyset$
5: **while** $|\mathcal{E}| \leq 0.185 \times |\mathcal{D}|$ **do**
6:     Add to $\mathcal{E}$ first element of U
7:     Remove first element from U
8: **end while**
9: $\mathcal{T} \leftarrow U$
10: **return** $\mathcal{T}, \mathcal{E}$

---

## C MOLECULAR FINGERPRINTS AND PHYSICOCHEMICAL DESCRIPTORS

**ECFP (Extended-connectivity fingerprints)** (Rogers & Hahn, 2010) are binary fingerprints where an "on" bit represents the presence of a substructure and an "off" bit, its absence. The radius determines the number of hops consider in the molecular graph to define each substructure, whereas the number of bits determines the level of information compression applied with smaller numbers leading to higher compression and increasing the likelihood of "collisions", i.e., two substructures being assigned the same bit. We have chosen the values of radius 2 and 2,048 bits as it is their most common configuration (Rogers & Hahn, 2010; Orsi & Reymond, 2024) and an early exploratory analysis showed that they were the most cost effective configuration.

**MACCS (Molecular ACCess System)** (Durant et al., 2002) are binary fingerprints, as well, with 166 bits. They differ from ECFPs in that there is a direct mapping between the bits and specific, pre-defined, chemical substructures.

**MAPc (MinHashed Atom-Pair chiral fingerprint)** (Orsi & Reymond, 2024) fingerprints are hash-based fingerprints, as ECFP, but they are not binary, each position can be occupied by different hash-values. They have been shown to outperform any other fingerprint in similarity searches with unique properties such as being able to discriminate between different stereoisomers of the same molecule. Number of bits and radius was selected based on the best configuration reported by its authors.

The **Lipinski** physicochemical descriptors are a vector that we introduce in this study for the purposes of measuring molecular similarity that has been inspired by Lipinski's rule of 5 (Lipinski et al., 1997). It describes the following properties of a molecule: 1) number of H acceptors, 2) number of H donors, 3) number of heteroatoms, 4) number of rotable bonds, 5) number of saturated carbocycles, 6) number of rings. The calculation of each the properties was performed through the implementation in RDKit (Landrum, 2013).

The **BUTC** is a vector with 8 terms describing the lower and upper bounds of the confidence interval of the following molecular characteristics related to the maximum value of the Burden Cluster-based Van der Waals atomic surface area (Stanton, 1999): 1 and 2) upper and lower bound of the mass eigenvalue; 3 and 4) upper and lower bound of the eigenvalue of the Gasteiger chargea; 5 and 6) upper and lower bound of the eigenvalue for Crippen logP; and 7 and 8) upper and lower bound of the eigenvalue for molar refractivity. It is calculated through the implementation in RDKit(Landrum, 2013).

## D  CHOICE OF SIMILARITY FUNCTION FOR THE ADMET DATASETS

Figure S1 displays detailed results per dataset and Tables S1-2 contain the raw data from each of the experiments. It is clear that similarity functions with small GOOD curve dynamic ranges, like all involving MACCS fingerprints or the BUTC descriptors, also demonstrate the biggest variance in their GOOD curve monotonicity. This is completely reasonable, as the smaller the dynamic range, the smaller the number of points comprising the GOOD curve and the more sensitive that the monotonicity calculation will be to random noise.

For the **Ames'** mutagenicity test dataset the similarity function with the largest dynamic range is Molformer (0.85), however this similarity function is ranked 9th in monotonicity ($0.24 \pm 0.02$). The next largest dynamic range belongs to MAPc - Jaccard (0.75), which is also the highest monotonicity ($0.82 \pm 0.01$). The GOOD curve corresponding to MAPc - Jaccard shows three phases. First, an approximately stationary phase at thresholds $0.2 - 0.45$, then a phase or steep growth $0.45 - 0.60$ and, finally, a phase with a more controlled growth $0.60 - 0.95$. It is also worth noting that for this dataset all the similarity functions with a dynamic range below 0.4 (with the exception of RG - ECFP2), have either a high variance on their monotonicity or very low monotonicity. This observation suggests that even for such a large dataset ($\approx 7,000$ molecules), the increased number of thresholds is necessary to avoid random noise.

For the **Drug-Induced Liver Injury**, DILI, dataset the two largest dynamic ranges are Molformer and MAPc - Jaccard (0.85), with MAPc - Jaccard ranked 3rd in monotonicity ($0.81 \pm 0.02$) and Molformer 16th ($0.33 \pm 0.02$). The GOOD curve of MAPc-Jaccard shows only one phase of steady growth. Interestingly, for this small dataset (476 molecules), the GOOD curve is less smooth than in the rest of the datasets, with small jumps between one threshold and the next. However, this jumps are well within the error ranges and the overall trend of the curve is clearly monotonic. Another interesting aspect of this dataset is that all similarity functions show decent monotonicity (lowest is $0.33 \pm 0.02$), which may respond to the molecular diversity within the dataset.

The **PAMPA NCATS** dataset is the most challenging of all the datasets considered in this study. Only the Molformer similarity function achieves a positive monotonicity, albeit a small one ($0.13 \pm 0.02$). We hypothesize that this is a consequence of the highly unbalanced nature of the dataset (1,739 positives to 295 negatives), which is further exacerbated by CCPart. Another contributing factor to this behaviour is the nature of the chemical predictive task, cell permeability, which is not mediated by the scaffold of the molecule or its overall structure, but for smaller local properties like the charges at the ends of the molecule or its lipophilicity (Hannesschlaeger et al., 2019). This is also shown in the Caco2 dataset (see below), which also shows uncharacteristic low monotonicity for fingerprint-based similarity metrics. Overall, this dataset shows the importance of working with good quality datasets and the need for further research into similarity functions within the biochemical domain.

Similarly, the **Caco2 cell effective permeability** dataset shows that MAPc-Jaccard and Molformer are tied as the similarity functions with largest dynamic range (0.85), but Molformer has the 2nd best monotonicity ($0.76 \pm 0.03$) and MAPc - Jaccard the 5th ($0.45 \pm 0.02$). Interestingly the 1st similarity function in monotonicity is ChemBERTa-2 ($0.80 \pm 0.05$), which is also based on a Language Model.

These results are consistent with those seen for the PAMPA dataset, and indicate some dependence on the chemical task (in both cases cell penetration). If we look at the GOOD curve for Molformer, even though monotonic, most of the variance due to the threshold is concentrated at very low thresholds (0.1 - 0.25) and then the growth of the curve is very modest. It is clear from this observation and the previous one with PAMPA that there isi a need for further research into similarity functions that do not focus as much in global molecular structure (like common fingerprints), but also on the physicochemical characteristics of the molecules.

The **Half-life** dataset is quite idiosyncratic, as well. The three similarity functioons with the largest dynamic range (Molformer, 0.85; MAPc - Jaccard, 0.80; and Sokal - ECFP2, 0.70) all show monotonicity below or around 0.0. The fourth largest dynamic range corresponds to the Lipinski similarity function, which is a new physicochemical similarity function that we first introduced in this study to address the shortcomings of other alternatives. It has a decent, albeit small, monotonicity $(0.46 \pm 0.06)$. Interestingly, the best monotonicity is achieved by the other physicochemical similarity function we have introduced in this study (BUTC) with $0.80 \pm 0.06$, however, it has two short a dunamimc range (0.25). The Lipinski GOOD curve shows a some noise at the beginnig, probably caused by a pathological sampling by CCPart at the lowest threshold; and then a steep growth (0.40 - 0.50). From 0.50 until the end model performance remains static. The dataset is quite small ($\approx 667$ molecules) and this behaviour could be explained by a lack of internal chemical diversity. Alternatively, the Lipinski similarity function might not be fine-grained enough to properly exploit the existing diversity in the dataset. Overall, the results from this dataset paired with the previous two datasets highlight the importance of considering more diverse similarity functions that do not rely on the traditional hash-based fingerprints; and the need within the field for more research into other types of similarity functions.

Finally, the **LD50** dataset shows a similar behaviour to that of Ames, which may be explained by both of them being the biggest. Interestingly, Molformer shows a larger dynamic range (0.85) than MAPc - Jaccard (0.70); however Molformer ranks 12th in monotonicity $(0.48 \pm 0.02)$ and MAPc - Jaccard, 6th $(0.94 \pm 0.00)$, but has an almost perfect monotonicity.

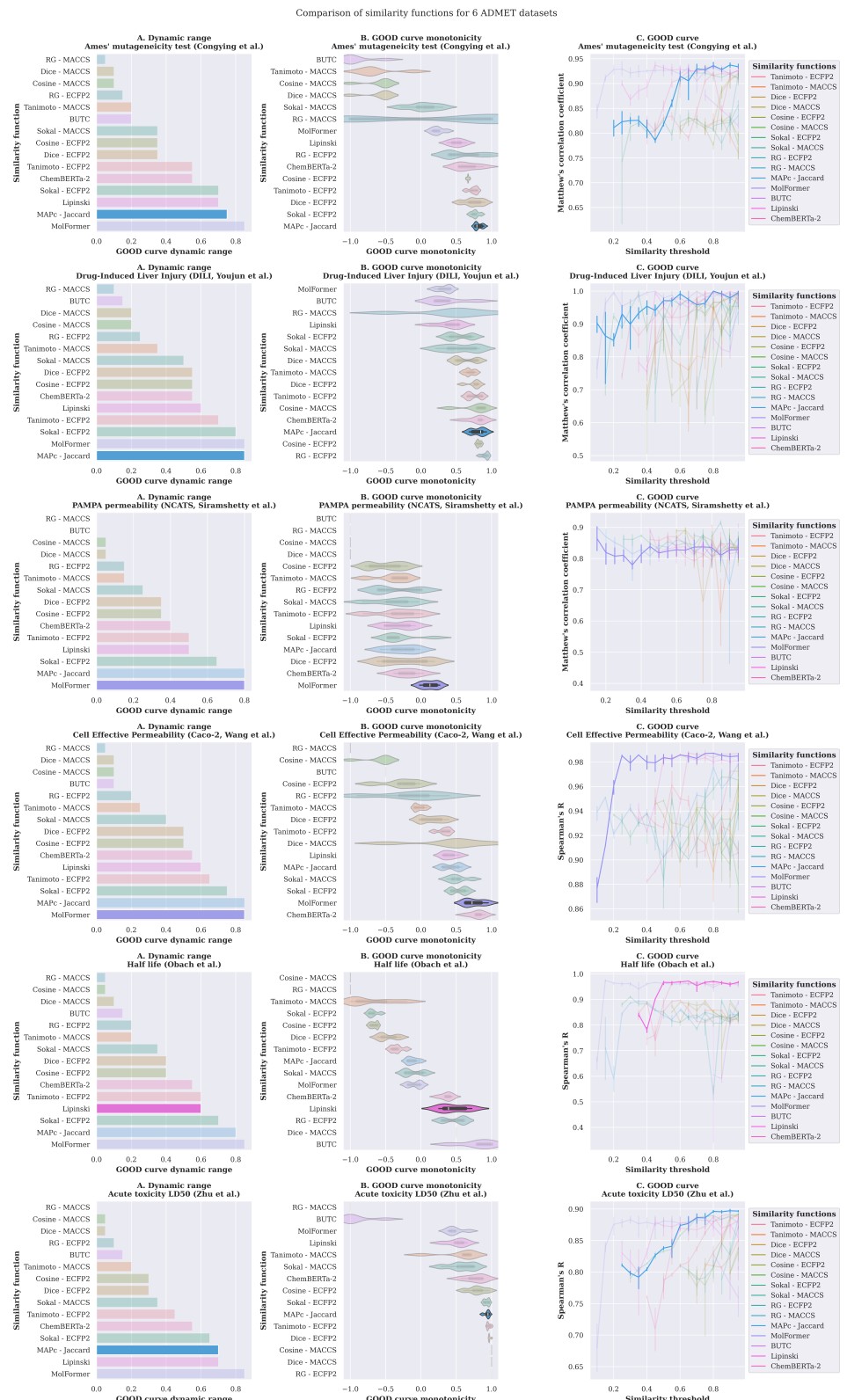

Figure 1: A: Analysis of dynamic range. B: Analysis of monotonicity, violin plot show the dispersion across 5 runs. C: GOOD curves, error bars correspond to the standard error of the mean across 5 runs with independent hyperparameter optimisation. The similarity metric chosen for each dataset is highlighted.

Table 1: Complete analysis of all similarity functions (Part I). Measurement error corresponds to the standard error of the mean for 5 independent runs.

| Dataset | Similarity metric | Monotonicity | Dynamic range |
|---|---|---|---|
| ames | Tanimoto - ECFP2 | $0.74 \pm 0.00$ | 0.55 |
| ames | Tanimoto - MACCS | $-0.62 \pm 0.01$ | 0.20 |
| ames | Dice - ECFP2 | $0.74 \pm 0.01$ | 0.35 |
| ames | Dice - MACCS | $-0.60 \pm 0.01$ | 0.10 |
| ames | Cosine - ECFP2 | $0.67 \pm 0.00$ | 0.35 |
| ames | Cosine - MACCS | $-0.60 \pm 0.01$ | 0.10 |
| ames | Sokal - ECFP2 | $0.77 \pm 0.00$ | 0.70 |
| ames | Sokal - MACCS | $0.03 \pm 0.02$ | 0.35 |
| ames | RG - ECFP2 | $0.60 \pm 0.02$ | 0.15 |
| ames | RG - MACCS | $0.20 \pm 0.10$ | 0.05 |
| ames | **MAPc - Jaccard** | $0.82 \pm 0.01$ | 0.75 |
| ames | MolFormer | $0.24 \pm 0.01$ | 0.85 |
| ames | BUTC | $-0.90 \pm 0.04$ | 0.20 |
| ames | Lipinski | $0.51 \pm 0.02$ | 0.70 |
| ames | ChemBERTa-2 | $0.66 \pm 0.07$ | 0.55 |
| caco2 | Tanimoto - ECFP2 | $0.33 \pm 0.00$ | 0.65 |
| caco2 | Tanimoto - MACCS | $-0.02 \pm 0.00$ | 0.25 |
| caco2 | Dice - ECFP2 | $0.16 \pm 0.01$ | 0.50 |
| caco2 | Dice - MACCS | $0.40 \pm 0.02$ | 0.10 |
| caco2 | Cosine - ECFP2 | $-0.28 \pm 0.01$ | 0.50 |
| caco2 | Cosine - MACCS | $-0.60 \pm 0.01$ | 0.10 |
| caco2 | Sokal - ECFP2 | $0.55 \pm 0.01$ | 0.75 |
| caco2 | Sokal - MACCS | $0.52 \pm 0.01$ | 0.40 |
| caco2 | RG - ECFP2 | $-0.14 \pm 0.04$ | 0.20 |
| caco2 | RG - MACCS | $-1.00 \pm 0.00$ | 0.05 |
| caco2 | MAPc - Jaccard | $0.45 \pm 0.01$ | 0.85 |
| caco2 | **MolFormer** | $0.76 \pm 0.02$ | 0.85 |
| caco2 | BUTC | $-0.50 \pm 0.00$ | 0.10 |
| caco2 | Lipinski | $0.41 \pm 0.02$ | 0.60 |
| caco2 | ChemBERTa-2 | $0.80 \pm 0.05$ | 0.55 |
| dili | Tanimoto - ECFP2 | $0.75 \pm 0.00$ | 0.70 |
| dili | Tanimoto - MACCS | $0.69 \pm 0.00$ | 0.35 |
| dili | Dice - ECFP2 | $0.75 \pm 0.00$ | 0.55 |
| dili | Dice - MACCS | $0.62 \pm 0.01$ | 0.20 |
| dili | Cosine - ECFP2 | $0.82 \pm 0.00$ | 0.55 |
| dili | Cosine - MACCS | $0.76 \pm 0.01$ | 0.20 |
| dili | Sokal - ECFP2 | $0.54 \pm 0.01$ | 0.80 |
| dili | Sokal - MACCS | $0.54 \pm 0.02$ | 0.50 |
| dili | RG - ECFP2 | $0.91 \pm 0.00$ | 0.25 |
| dili | RG - MACCS | $0.40 \pm 0.05$ | 0.10 |
| dili | **MAPc - Jaccard** | $0.81 \pm 0.01$ | 0.85 |
| dili | MolFormer | $0.33 \pm 0.01$ | 0.85 |
| dili | BUTC | $0.36 \pm 0.04$ | 0.15 |
| dili | Lipinski | $0.41 \pm 0.04$ | 0.60 |
| dili | ChemBERTa-2 | $0.80 \pm 0.05$ | 0.55 |
| pampa ncats | Tanimoto - ECFP2 | $-0.33 \pm 0.01$ | 0.50 |
| pampa ncats | Tanimoto - MACCS | $-0.40 \pm 0.01$ | 0.15 |
| pampa ncats | Dice - ECFP2 | $-0.22 \pm 0.01$ | 0.35 |
| pampa ncats | Dice - MACCS | $-1.00 \pm 0.00$ | 0.05 |
| pampa ncats | Cosine - ECFP2 | $-0.49 \pm 0.01$ | 0.35 |
| pampa ncats | Cosine - MACCS | $-1.00 \pm 0.00$ | 0.05 |
| pampa ncats | Sokal - ECFP2 | $-0.28 \pm 0.02$ | 0.65 |
| pampa ncats | Sokal - MACCS | $-0.36 \pm 0.02$ | 0.25 |
| pampa ncats | RG - ECFP2 | $-0.36 \pm 0.02$ | 0.15 |
| pampa ncats | RG - MACCS | $-1.00 \pm 0.00$ | 0.00 |
| pampa ncats | MAPc - Jaccard | $-0.27 \pm 0.02$ | 0.80 |
| pampa ncats | **MolFormer** | $0.13 \pm 0.02$ | 0.80 |
| pampa ncats | BUTC | $-1.00 \pm 0.00$ | 0.00 |
| pampa ncats | Lipinski | $-0.30 \pm 0.05$ | 0.50 |
| pampa ncats | ChemBERTa-2 | $-0.18 \pm 0.07$ | 0.40 |
| half life | Tanimoto - ECFP2 | $-0.34 \pm 0.00$ | 0.60 |
| half life | Tanimoto - MACCS | $-0.70 \pm 0.01$ | 0.20 |
| half life | Dice - ECFP2 | $-0.47 \pm 0.01$ | 0.40 |
| half life | Dice - MACCS | $0.50 \pm 0.00$ | 0.10 |
| half life | Cosine - ECFP2 | $-0.67 \pm 0.00$ | 0.40 |
| half life | Cosine - MACCS | $-1.00 \pm 0.00$ | 0.05 |
| half life | Sokal - ECFP2 | $-0.68 \pm 0.00$ | 0.70 |
| half life | Sokal - MACCS | $-0.11 \pm 0.01$ | 0.35 |
| half life | RG - ECFP2 | $0.48 \pm 0.01$ | 0.20 |
| half life | RG - MACCS | $-1.00 \pm 0.00$ | 0.05 |
| half life | MAPc - Jaccard | $-0.12 \pm 0.01$ | 0.80 |
| half life | MolFormer | $-0.08 \pm 0.01$ | 0.85 |
| half life | BUTC | $0.80 \pm 0.04$ | 0.15 |
| half life | **Lipinski** | $0.46 \pm 0.05$ | 0.60 |
| half life | ChemBERTa-2 | $0.36 \pm 0.04$ | 0.55 |

Table 2: Complete analysis of all similarity functions (Part II). Measurement error corresponds to the standard error of the mean for 5 independent runs.

| Dataset | Similarity metric | Monotonicity | Dynamic range |
|---------|-------------------|--------------|---------------|
| ld50 | Tanimoto - ECFP2 | $0.96 \pm 0.00$ | 0.45 |
| ld50 | Tanimoto - MACCS | $0.54 \pm 0.01$ | 0.20 |
| ld50 | Dice - ECFP2 | $0.97 \pm 0.00$ | 0.30 |
| ld50 | Dice - MACCS | $1.00 \pm 0.00$ | 0.05 |
| ld50 | Cosine - ECFP2 | $0.79 \pm 0.01$ | 0.30 |
| ld50 | Cosine - MACCS | $1.00 \pm 0.00$ | 0.05 |
| ld50 | Sokal - ECFP2 | $0.93 \pm 0.00$ | 0.65 |
| ld50 | Sokal - MACCS | $0.57 \pm 0.01$ | 0.35 |
| ld50 | RG - ECFP2 | $1.00 \pm 0.00$ | 0.10 |
| ld50 | RG - MACCS | $-1.00 \pm 0.00$ | 0.00 |
| ld50 | **MAPc - Jaccard** | $0.94 \pm 0.00$ | 0.70 |
| ld50 | MolFormer | $0.48 \pm 0.02$ | 0.85 |
| ld50 | BUTC | $-0.90 \pm 0.05$ | 0.15 |
| ld50 | Lipinski | $0.51 \pm 0.06$ | 0.70 |

# E HYPERPARAMETER SEARCH FOR HPO

Table S3 describes the hyperparameter space defined for all experiments.

Table 3: Hyperparameter search space for each learning algorithm.

| Model | Trials | Hyperparameter search space | | | |
|-------|--------|------|------|-------|-----------|
| | | Name | Type | Range | Log-scale |
| SVM | 200 | C | float | $1 \times 10^{-3}$ - $10^3$ | Yes |
| | | kernel | categorical | linear, poly, rbf, sigmoid | NA |
| | | degree (only kernel poly) | integer | 2-5 | No |
| | | coef0 (only with poly or sigmoid) | float | $10^{-8}$ - 1 | Yes |
| | | epsilon (only regression) | float | $10^{-5}$ - 1 | Yes |
| KNN | 200 | K | integer | 1-30 | No |
| | | Weights | categorical | uniform, distance | NA |
| | | algorithm | categorical | ball tree, kd tree, brute | NA |
| | | leaf size (only with ball or kd tree) | integer | 5 - 100 | No |
| | | power of Minkowski metric | integer | 1 - 3 | No |
| RF | 200 | number of estimators | integer | 10-5,000 | No |
| | | criterion | categorical | gini*, entropy*, log loss*, MSE**, MAE**, friedman MSE** | NA |
| | | minimum samples per split | integer | 2 - 100 | No |
| | | maximum features | categorical | log2, sqrt | NA |
| | | complexity parameter (ccp_alpha) | float | $10^{-10}$ - $10^{-3}$ | Yes |
| LightGBM | 200 | number of estimators | integer | 10-5,000 | No |
| | | minimum child samples | integer | 10 - 500 | No |
| | | minimum split gain | float | $10^{-10}$ - $10^{-3}$ | Yes |
| | | regularization $\alpha$ | $10^{-10}$ - $10^{-3}$ | Yes | |
| | | learning rate | float | $10^{-7}$ - $10^{-1}$ | Yes |

# F VISUALISATION OF THE CHEMICAL SPACE

This appendix contains the visualisation of the level of overlap between the training and testing partitions of the dataset in the chemical space defined by the ECFP fingerprints.

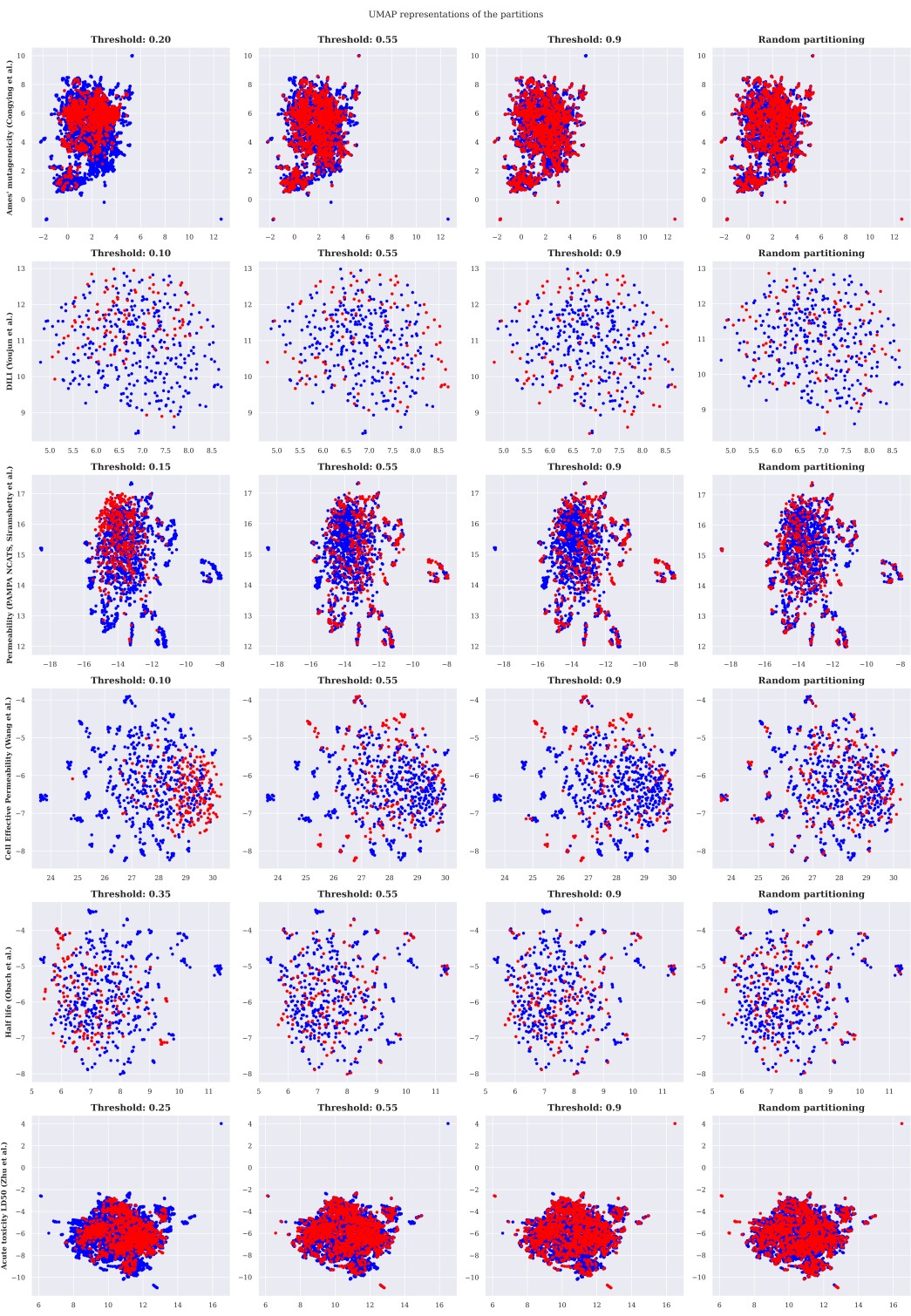

Figure 2: UMAP representation of the chemical space covered by the training (blue) and testing (red) partitions of all molecular datasets considered during this study.

## G    DETAILED HESTIA ANALYSES FOR ALL ADMET DATASETS

This appendix contains the detailed Hestia analyses for all ADMET datasets considered in this study.

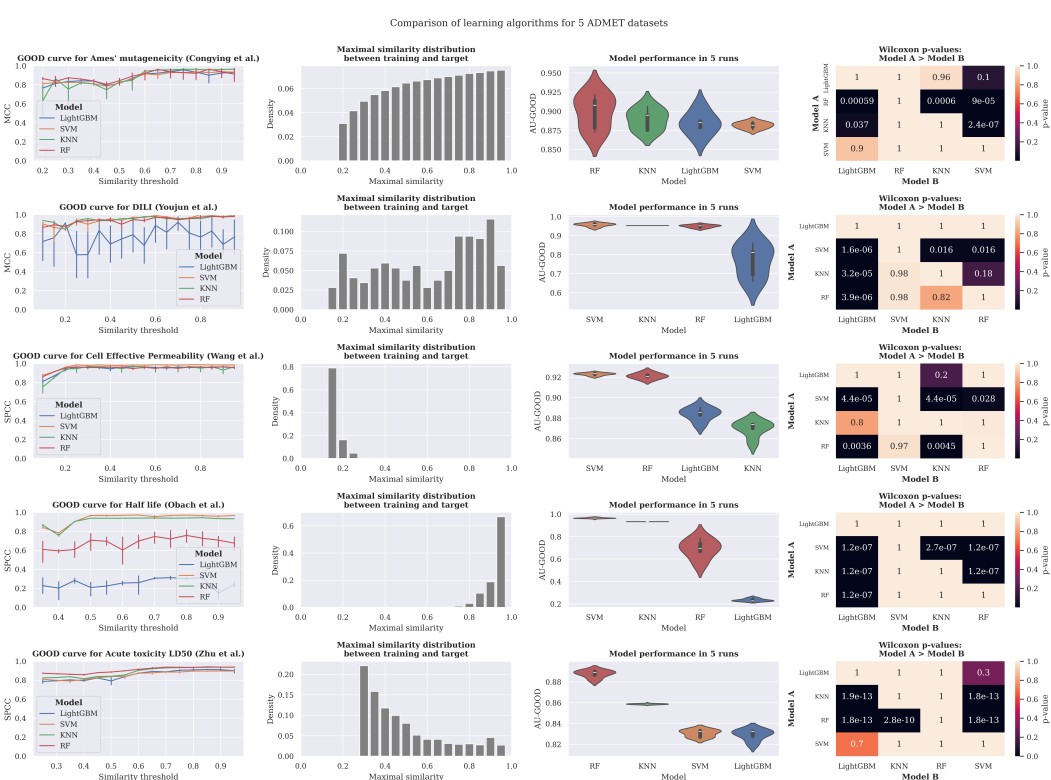

Figure 3:    Hestia analyses for the comparison of learning algorithms for the 6 ADMET datasets considered in this study.

## H    DETAILED HESTIA ANALYSES FOR ALL PROTEIN DATASETS

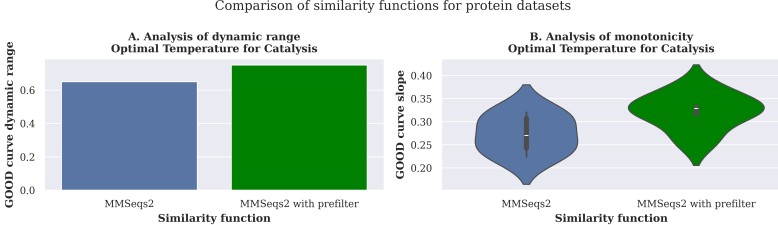

Figure 4: A: Analysis of dynamic range. B: Analysis of monotonicity. Violin plot show the dispersion in the GOOD curve slope across 8 alternative representation methods and 5 runs per representation (total of 40 experiments)

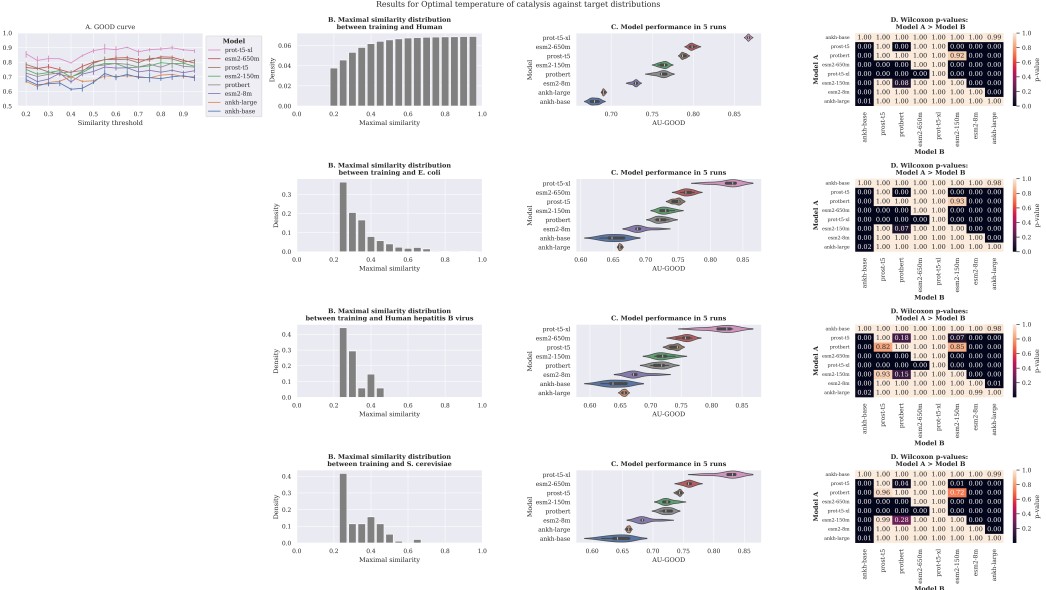

Figure 5: Analyses for the comparison of PLM embeddings algorithms for the optimal temperature for catalysis.

