# OpenReview forum: "A new framework for evaluating model out-of-distribution generalisation for the biochemical domain"
_ICLR.cc/2025/Conference — ICLR 2025 Poster_

### Official Review · Reviewer_mZMB · 2024-11-03

**Soundness:** 2
**Presentation:** 2
**Contribution:** 1
**Rating:** 6
**Confidence:** 3

**Summary:**

This paper introduces a formalized approach for creating training and test splits that assesses the generalization performance of a machine learning algorithm on biological data as a function of a (chosen) similarity metric between data points.

**Strengths:**

* The paper is generally well written.
* It is broadly applicable to a wide range of biological domains given the generality of the similarity metric based approach.

**Weaknesses:**

* The paper is written in an overly formalized manner. Sections 2.1 and 2.2 say very little that’s non-obvious. It can be described in 1-2 paragraphs of plain English. The over-formalization of very simple concepts seem largely intended to make the matter seem more complex than it actually is. Similarly for sec 2.3, which introduces more novel methods and is not strictly a pedagogical section, but the description is far too long and unnecessarily mathematical.
* While the basic concepts are overly formalized, concepts that could actually use more formal precision, e.g., the behavior of different GOOD curves, are instead described using phrases like “visually incoherent GOOD curves” without actually specifying what “visually incoherent” means.
* In most biological domains, data points are evolutionarily related (e.g. proteins, species, etc), and so the assumption that there will be disconnected subgraphs is not really correct, at least not for stringent levels of generalization. I.e., it might be possible to find a threshold by which the graph does become disconnected, but that threshold would only segregate highly similar data points and would result in a very easy test.
* In many biological domains, data points are noisy, and it’s quite possible that the type of disconnected subgraphs found in the proposed algorithm would basically correspond to pathological outliers. Thus I would caution against naive application of this approach as it may result in test sets comprising largely unrepresentative data points.
* Ultimately there’s little utility / novelty here. The approach does not introduce anything substantially different from the existing literature, and in the places where it does differ, it is hard to judge whether this approach will actually work in practice. The experiments performed are not compared to any other than baseline and the discussion about the resulting GOOD curves is largely informal and speculative.

**Questions:**

* I would ask the authors to think of ways to demonstrate that their approach outperforms other existing methods.

---

> ### Author Response · Authors · 2024-11-21
> **Response to Reviewer mZMB**
>
> We thank Reviewer mZMB for their criticism of our manuscript, as it has given us the opportunity to improve the quality of our analysis and the overall text.
>
> ---
>
> **Q1. The paper is written in an overly formalized manner [...].**
>
> **A1.** Thank you for this observation. We agree that the formalism in Sections 2.1–3 could make reading these sections challenging and appreciate the suggestion to include “plain English” explanations. To address this, we have revised the manuscript to prioritize accessibility, moving formal mathematical definitions to Appendices A and B, and replacing them in the main text with more concise, intuitive descriptions.
>
> **Q2. [...], concepts that could actually use more formal precision, e.g., the behavior of different GOOD curves, are instead described using phrases like “visually incoherent GOOD curves” without actually specifying what “visually incoherent” means.**
>
> **A2.** We appreciate your concern about the lack of formal precision in the discussion of the results. We have identified and dropped ambiguous phrases from the text like *“visually incoherent”*, *“notably different shapes”*, and *“the relationship between similarity threshold and model performance changes”*. We have also improved the precision of our analyses throughout the main text and included in-depth analyses for each dataset in Appendix D, to provide a more detailed description of our findings and their interpretation.
>
> Particularly, regarding the analysis of the *“visual incoherent GOOD curves”*, we now use Spearman’s correlation coefficient to objectively quantify the monotonicity of the curves.
>
> **Q3. In most biological domains, data points are evolutionarily related (e.g. proteins, species, etc), and so the assumption that there will be disconnected subgraphs is not really correct, at least not for stringent levels of generalization. [...].**
>
> **A3.** Thank you for highlighting this concern. We agree that traditional similarity metrics, such as those based on global features (e.g., structure-based fingerprints or sequence identity), may fail to create meaningful partitions in scenarios where biological data points are closely related. To address this issue we discuss a dataset, where this issue occurs and the similarity metrics that could be used to define meaningful generalization, in Section 3.2, as follows:
>
> *”However, for other protein tasks where data points are heavily evolutionarily related we recommend using alternative metrics like the Hamming distance used in the Fluorescence task in the TAPE benchmarks (Rao et al., 2019) or the distance in a phylogenetic tree.”*
>
> Further, we also include a remark in the conclusions, to clarify that our framework is built to assist in the development and evaluation of such custom similarity metrics:
>
> *”We provide a robust framework for analysing whether a similarity function is appropriate for partitioning a given dataset, with a series of quantitative metrics to provide a formal analysis of the reliability of the resulting GOOD curves.”*

---

> ### Author Response · Authors · 2024-11-21
>
> **Q4. In many biological domains, data points are noisy, and it’s quite possible that the type of disconnected subgraphs found in the proposed algorithm would basically correspond to pathological outliers. Thus I would caution against naive application of this approach as it may result in test sets comprising largely unrepresentative data points.**
>
> **A4.** We agree that this approach should not be applied naively, and included in Subsection 3.1.1. tests to diagnose any possible problems:
>
> *”A necessary assumption of the GOOD and AU-GOOD calculations is that model performance can be described as a function of the similarity threshold. Therefore, Spearman's correlation coefficient serves as a robust, non-parametric diagnostic test to ensure that any GOOD curve meets this assumption, without assuming linearity.”*
>
> And:
>
> *”It also serves as a diagnostic test to ensure that the GOOD curves meet the assumption of model performance dependence on the similarity thresholds. If none of the similarity functions achieves a reasonable monotonicity, it is likely that a custom similarity function is required; or that there are underlying issues with the data generation process that need to be corrected in a dataset specific way”*
>
> We also include further discussion in Appendix A.3, as follows:
>
> *”In many biological domains, data points are noisy, and it’s quite possible that the type of disconnected subgraphs found in the proposed algorithm would bias the testing set towards unrepresentative regions of the (bio)chemical space. This behaviour would lead to unstable GOOD curves where there is low monotonic correlation between similarity threshold and model performance and would therefore be identified when following the guidelines outlined in Subsection 3.1.1.*
>
> *However, if the GOOD curve is monotonic, there could be some specific partitions at given thresholds that show unstable behaviour. Given that the AU-GOOD is an expected value, with a sufficient number of thresholds (n > 10) and a wide enough dynamic range, the estimated and the true population AU-GOOD values will tend to converge.”*
>
> Finally, we discuss the case of PAMPA, which is a dataset where the behaviour described appears and is properly diagnosed by our diagnostic test. We include this text in Subsection 3.1.1, as follows:
>
> *”It is worth noting that the PAMPA dataset has low monotonicity with all similarity metrics considered in this study, which makes the interpretation of the GOOD curve unreliable. We believe that this behaviour is due to the idiosyncrasies of this dataset which is relatively small ($\approx2,000$ molecules) and highly unbalanced (1,739 positives to 295 negatives). The imbalance is further exacerbated by the partitioning algorithm with the test set containing, in most partitions, around two thirds of all negatives $\approx 200$. This dataset provides a concrete showcase of the limitations of our framework, which are: 1) it depends on the similarity metrics and for certain tasks it may be needed to define custom similarity functions and 2) it may not be reliable for highly unbalanced datasets as the partitioning can increase the imbalance. For this reason, the PAMPA dataset will be excluded from the next step, as the GOOD curve and AU-GOOD values are not reliable.*
>
> *In any case, we highly recommend performing an initial analysis with any similarity function to determine whether the GOOD has an appropriate dynamic range and meets the assumption of correlation between similarity threshold and model performance.”*

---

> ### Author Response · Authors · 2024-11-21
>
> **Q5. Ultimately there’s little utility / novelty here. The approach does not introduce anything substantially different from the existing literature, and in the places where it does differ, it is hard to judge whether this approach will actually work in practice. The experiments performed are not compared to any other than baseline and the discussion about the resulting GOOD curves is largely informal and speculative.**
>
> **A5.** We sincerely appreciate the honesty of your assessment. We have introduced several improvements to the manuscript to try to properly highlight the novel aspects of our work.
>
> First, regarding the **novelty of the AU-GOOD metric** in Section 2.1, as follows:
>
> *”This is a significant advantage when compared with other approaches for out-of-distribution performance estimation, like MOOD (Tossou et al., 2024) which focuses on generating test sets that closely resemble the target deployment distribution and requires repeating the training/evaluation cycle for each new target deployment distribution; whereas with ours, only the similarity histogram needs to be updated. Our approach is also similar to Ektefaie et al. (2024) with three main differences: 1) their method defines arbitrarily a certain similarity threshold and considers model performance against the proportion of similar entities between training and testing; whereas, our approach defines a certain proportion of similar entities ($0\%$) and explores the effect on model performance of the different similarity thresholds; 2) their method assumes that whatever similarity function they have chosen is good at capturing the relevant properties that mediate any given task; however, our method makes no such assumption and we provide a series of analyses to determine that a GOOD curve reasonably describes out-of-distribution generalisation; and 3) their approach does not provide any estimation of model performance conditioned on the target deployment distribution(s), which limits its practical application in real-world scenarios.”*
>
> Regarding the **novelty of the CCPart algorithm** in Section 2.2, as follows:
>
> *”There are three conditions that the partitioning algorithms have to fullfil are: 1) to not remove any data point, otherwise the comparison between thresholds could also correspond to the loss of information and would make them not directly comparable; 2) to enforce strict boundaries between training and testing subsets for any given similarity threshold (Walsh et al., 2020), otherwise we would not be measuring strictly the dependence on the similarity threshold, but also (implicitly) the effect of overlap within any given threshold; and 3) to allow for multiple thresholds, otherwise we cannot describe model performance as a function of the threshold.*
>
> *All prior partitioning algorithms violate at least one of the three conditions: GraphPart (Teufel et al., 2023) or Lo-Hi splitting (Steshin, 2023) do not fulfill condition 1; perimeter split, maximum dissimilarity split (Ramsundar et al., 2019), the Butina split (Butina, 1999), Dyonisus (Tom et al., 2023), and spectra (Ektefaie et al., 2024) do not fulfill condition 2; finally, scaffold split does not fulffil condition 3.”*
>
> Regarding the **novelty of using the GOOD curve to evaluate whether a similarity function is appropriate for guiding the partitioning of a certain dataset**, in the Introduction, as follows:
>
> *”In contrast to previous methods that choose a certain similarity function assuming that it can capture generalisation within the context of a certain modelling task (Butina, 1999; Ramsundar et al., 2019; Tom et al., 2023; Steshin, 2023; Teufel et al., 2023; Tossou et al., 2024; Ektefaie et al., 2024), we propose a framework that allows to quantify how appropriate a similarity metric is for guiding the partitioning of a certain dataset, with minimal relience on domain knowledge.“*
>
> And in Section 3.1.1:
>
> *”The main conclusion that can be drawn from these results is that there is no one-size-fits-all similarity function. Different datasets require different similarity metrics. This indicates that before starting any analysis regarding model generalisation, a careful selection of the similarity metric used to measure generalisation is crucial. This initial step of analysis is missing from previous works (Butina, 1999, Ramsundar et al., 2019; Steshin, 2023; Tom et al., 2023; Tossou et al., 2024; Ektefaie et al., 2024).”*
>
> Regarding the **formal analysis of the AU-GOOD**, we refer to Appendix A, where the mathematical properties of the metric are described.

---

> ### Author Response · Authors · 2024-11-21
>
> **Q6. I would ask the authors to think of ways to demonstrate that their approach outperforms other existing methods.**
>
> **A6.** The reviewer rightly identifies this as an important question, which we have been thinking about. However, for none of the biological benchmarks available to us is there such an external ground truth.  To clarify this issue with method evaluation we include the following clarification in Section 2.1:
>
> *”Finally, our study is limited by the lack of experimental follow up studies to confirm empirically the advantages of our method. The closest study to ours in this regard are MOOD from (Tossou et al., 2024) and Spectra (Ektefaie et al., 2024), neither of which provided a metric to evaluate the approach on “ground truth” data. Development of scenarios where such a ground truth exists will help inform evaluation of the performance of our method.”*
>
> And with regards to the partitioning algorithms, in Section 2.2:
>
> *”In any case, our framework is agnostic to the specific partitioning strategy used and future development of alternative algorithms that fulfill the three conditions could improve the accuracy of the AU-GOOD calculation. It is also an open research question what the impact is of violating any of the three assumptions on the reliability of the AU-GOOD metric.”*
>
> - - -
>
> We sincerely thank the reviewer for their insightful and frank comments as we believe they have contributed to a major improvement in the quality of the revised manuscript.

---

> > ### Comment · Reviewer_mZMB · 2024-11-25
> >
> > Thanks for addressing my concerns. You have substantially revised the text to increase its clarity and to highlight the novelty of your approach and where it is adding value. I have therefore increased my score to 6.

---

### Official Review · Reviewer_53DR · 2024-11-04

**Soundness:** 3
**Presentation:** 2
**Contribution:** 2
**Rating:** 6
**Confidence:** 4

**Summary:**

This paper proposes AU-GOOD, a metric for assessing machine learning model generalization to out-of-distribution biochemical data. AU-GOOD quantifies expected performance under increasing train-test set dissimilarity, integrating prior target distribution knowledge. A novel partitioning algorithm, CCPart, generates challenging test sets by prioritizing dissimilar data points. Statistical methods for model comparison based on AU-GOOD are also provided. The accompanying open-source code includes implementations of various similarity functions, partitioning algorithms, and AU-GOOD calculation, applied to two use cases: small molecule property prediction and protein language model embedding evaluation.

**Strengths:**

Authors presented a detailed analysis of AU-GOOD, and presented a CCPart for splitting sets based on pairwise similarities. CCPart can be applied to any dataset that has useful pairwise similarity metrics in the data, which means it can be applied to other datasets outside of biochemical applications. The authors perform a wide variety of tests on various datasets, and also study various protein language model embeddings, and their performances on OOD. The authors take care to use statistical tests to confirm statistical significance.

**Weaknesses:**

The font in the figures are extremely small. Please enlarge to improve readability. Especially the plots in the SI, ie. SI fig 1.

The use of pairwise similarity measures heavily restricts the size of the dataset that can be split. As such, most datasets the authors look at are on the smaller end. This also means that the models studied are usually traditional ML methods (KNN, SVM, RF, etc.).

Various works have already tried structure-only fingerprints-based splitting methods. This includes work done by the MOOD paper from Tossou et al. (2024), DeepChem from Ramsundar et al. (2019), and Dionysus from Tom et al. (2023). In particular, CCPart is quite similar to Butina or scaffold splitting (from DeepChem), which gathers clusters of fingerprints or scaffolds and sets a cutoff threshold for Tanimoto similarity.

Overall, the AUGOOD metric, and the GOOD curve, is not convincing to me as a reliable measure of OOD performance. Looking at the curves in SI Fig 2, the variations in the GOOD curve are not as well behaved as those in the main text, such as Fig 1, or Fig 5. Fitting a linear line to the GOOD curve to attain a "slope" also seems strange considering the huge variations in the performance metric. The curves also dramatically change as the performance metric is changed (ie. between Matthew vs Spearman vs Pearson coefficients). The error bars are very large on the performance metrics, but the authors seem to use the metrics of model performance interchangeably.

Additionally, the choice of AU the GOOD curve is confusing to me. Take for example, Fig 1A, where the authors conclude that Model A is less generalizable than B, because B performes similarily across the different similarity thresholds. However, the area under the curve would actually be quite similar for model A and B. While this is an example to illustrate the GOOD curve, it is also an example that shows how AUGOOD may not be a useful metric. The authors may defer to the slope of a linear fit, but as seen in Fig 2 of SI, the GOOD curve is rarely linear, or even monotonic.

**Questions:**

Are there any plans to extend the work to larger datasets and deep learning models? Often, datasets for virtual screening are much larger, on the order of millions, and GNN and GCN models start to exhibit state-of-the-art performance.

What motivates the use of various performance metrics? Why, for example, Spearman for Caco2, Halflife and LD509, but Matthew for Ames, DILI, and PAMPA? It seems that the behaviour of the GOOD curve is heavily dependent on this choice. Would the conclusions change for different metrics? And how might that affect the interpretation of AU-GOOD, if the metric can so quickly fluctuate and change?

How does CCPart compare with other splitting methods that are already available, such as Butina splits, scaffold splits, cluster splits etc.?

Table 1, I think the ranking should be 2,4,1,4 for the models read left to right?

---

> ### Author Response · Authors · 2024-11-21
> **Response to Reviewer 53DR**
>
> We thank Reviewer 53DR for their thoughtful comments and suggestions. We address your concerns in the following points:
>
> - - -
>
> **Q1. The font in the figures are extremely small. Please enlarge to improve readability. Especially the plots in the SI, ie. SI fig 1.**
>
> **A1.** We have increased the font size and have included supplemental tables that contain summary statistics from the Figures to facilitate their interpretation wherever there may be ambiguity (see Tables S1 and S2).
>
> **Q2. The use of pairwise similarity measures heavily restricts the size of the dataset that can be split. [...].**
>
> **A2.** We thank you for raising this issue as it allows us to discuss the practical limitations of our implementation. We have addressed it in Section 2.4, as follows:
>
> *”The partitioning algorithms rely on pairwise similarity calculations and scale in complexity and memory footprint by the $\mathcal{O}(N^2)$. In response to this, our implementation improves computational and memory efficiency by relying on sparse matrices and parallelism. We have run toy examples with datasets of up to 400,000 protein sequences within a day in 32 CPU cores and 200GB of RAM, thus we believe that scaling does not limit the usefulness of the method within the domain. In other domains, algorithmic improvements might be required for scaling to very large datasets.”*
>
> **Q3. Various works have already tried structure-only fingerprints-based splitting methods. [...]**
>
> **A3.** We appreciate you raising this issue, we have improved the comparison to the state-of-the-art. First, regarding the use of structure-only fingerprint-based similarity metrics, we have clarified that our work extends beyond those traditional methods in Section 3.1, as follows:
>
> *”We have also considered alternative similarity metrics that are not fingerprint-based, including: the euclidean distance between embeddings from two pre-trained chemical language models: Molformer (Ross et al., 2022) and ChemBERTa-2 (77M MLM) (Ahmad et al., 2022); the canberra distance between two types of vectors of physicochemical descriptors 1) BUTCD (Stanton, 1999) and Lipinski (Lipinski et al., 1997). More details in Appendix C.”*
>
> Second, regarding the novelty of the partitioning algorithm, we have clarified the comparison to the state-of-the-art in Section 2.2, as follows:
>
> *”The condition that the partitioning algorithm has to fullfil are: 1) to not remove any data point, otherwise the comparison between thresholds could also correspond to the loss of information and would make them not directly comparable; 2) to enforce strict boundaries between training and testing subsets for any given similarity threshold (Walsh et al., 2020), otherwise we would not be measuring strictly the dependence on the similarity threshold, but also (implicitly) the effect of overlap within any given threshold; and 3) to allow for multiple thresholds, otherwise we cannot describe model performance as a function of the threshold.*
>
> *All prior partitioning algorithms violate at least one of the three conditions: GraphPart (Teufel et al., 2023) or Lo-Hi splitting (Steshin, 2023) do not fulfill condition 1; perimeter split, maximum dissimilarity split (Ramsundar et al., 2019), the Butina split (Butina, 1999), Dyonisus (Tom et al., 2023), and Spectra (Ektefaie et al., 2024) do not fulfill condition 2; finally, scaffold split does not fulffil condition 3.”*
>
> We also compare to the work by Tossou et al. (2024), in Section 2.1, as follows:
>
> *”This is a significant advantage when compared with other approaches for out-of-distribution performance estimation, like MOOD (Tossou et al., 2024) which focuses on generating test sets that closely resemble the target deployment distribution and requires repeating the training/evaluation cycle for each new target deployment distribution; whereas with ours, only the similarity histogram needs to be updated.”*
>
> **Q4. Fitting a linear line to the GOOD curve to attain a "slope" also seems strange considering the huge variations in the performance metric.**
>
> **A4.** We agree with you that the “slope” was a gross approximation for the property that we should have been actually measuring, which is the monotonic correlation between threshold and model performance. Accordingly, we have substituted the slope as the second criterion for selecting an appropriate similarity metric for the monotonicity as defined by Spearman’s $\rho$ correlation coefficient. We justify this choice in Section 3.1.1, as follows:
>
> *”A necessary assumption of the GOOD and AU-GOOD calculations is that model performance can be described as a function of the similarity threshold. Therefore, Spearman's correlation coefficient serves as a robust, non-parametric diagnostic test to ensure that any GOOD curve meets this assumption, without assuming linearity.”*

---

> ### Author Response · Authors · 2024-11-21
>
> **Q5. Overall, the AUGOOD metric, and the GOOD curve, is not convincing to me as a reliable measure of OOD performance. [...]**
>
> **A5.** The results with the new similarity functions and the monotonicity as criterion for the suitability of any given GOOD curve to provide a reliable measure of OOD performance have greatly improved the curves in Figure S1. The results are summarised in the Table below, which corresponds to the new Table 1 in Section 3.1.1.
>
> | Dataset | Similarity metric | Dynamic range | Monotonicity |
> | --- | --- | --- |  --- |
> | Ames | MAPc - Jaccard | 0.75 | 0.82 ± 0.02 |
> | DILI | MAPc - Jaccard | 0.85 | 0.81 ± 0.02 |
> | PAMPA-NCATS | Molformer | 0.80 | 0.13 ± 0.02 |
> | Caco2 | Molformer | 0.85 | 0.76 ± 0.03 |
> | Half-life | Lipinski | 0.60 | 0.46 ± 0.06 |
> | LD50 | MAPc - Jaccard | 0.70 | 0.94 ± 0.02 |
>
> We have included an in–depth description of these results per individual dataset in Appendix D. and we further discuss the importance of choice of similarity function for the curve to be reliable in Section 3.1.1, as follows:
>
> *”The main conclusion that can be drawn from these results is that there is no one-size-fits-all similarity function. Different datasets require different similarity metrics. This indicates that before starting any analysis regarding model generalisation, a careful selection of the similarity metric used to measure generalisation is crucial. This initial step of analysis is missing from previous works (Butina, 1999, Ramsundar et al., 2019; Steshin, 2023; Tom et al., 2023; Tossou et al., 2024; Ektefaie et al., 2024).”*
>
> Regarding the error bars within each experiment, we have included the following clarification to the beginning of Section 3.1:
>
> *”Each run for a dataset and similarity metric has the exact same partitions, because the partitioning algorithm is heuristic and not stochastic, therefore any variance between runs depends on the learning algorithm and the Bayesian hyperparameter-optimisation which is conducted for each run independently with Optuna (Akiba et al., 2019) [...].”*
>
> **Q6. What motivates the use of various performance metrics? [...]**
>
> **A6.** We have clarified the text where we describe the rationale behind the selection of the performance metrics used, in the beginning of Section 3.1, as follows:
>
> *”We report model performance with Matthew's correlation coefficient for classification tasks (Ames, DILI, and PAMPA) and Spearman's $\rho$ correlation coefficient for regression tasks (Caco2, Half-life, and LD50).”*
>
> We believe that the variance observed in the original experiments was due to using the wrong similarity metrics when partitioning, and with the new results, it has been mitigated. Additionally, we include the following recommendation in Appendix A.4, regarding choice of performance metrics:
>
> *”We recommend the use of model performance metrics that are not sensitive to label imbalance like Matthew's correlation coefficient or weighted F1 for classification tasks and Spearman's or Pearson's correlation coefficient for regression tasks. Otherwise, the GOOD curve will be much more sensitive to certain partitions having test sets with different label distributions which could make the AU-GOOD calculation more unstable.”*

---

> ### Author Response · Authors · 2024-11-21
>
> **Q7. Additionally, the choice of AU the GOOD curve is confusing to me. [...]**
>
> **A7.** We thank you for raising this concern as it makes it clear that one of the core arguments of the paper was not clearly conveyed. We introduce a clarification for the usefulness of the AU-GOOD in the conclusion, as follows:
>
> *”The AU-GOOD is a new metric that estimates the expected model performance against any target deployment distribution(s), and thus provides a quantifiable value to guide the selection of the most appropriate model to use in different deployment scenarios, without the requirement for additional experiments.“*
>
> We also include a clarification on how it is calculated in Section 2, as follows:
>
> *”Given any number of arbitrary target deployment distribution(s), we calculate the similarity between each of its elements and the training data. For each element in the deployment distribution, we keep the maximum similarity and calculate a histogram of their distribution such that each bin corresponds to one of the thresholds in the GOOD curve. The AU-GOOD, or Area Under the GOOD curve, can be calculated as the weighted average of all points in the GOOD curve where the weight of each point corresponds to the value of the histogram for the corresponding threshold”*
>
> We clarify the fact that the AU-GOOD is designed to quantify the expected model performance relative to specific deployment scenarios, rather than as an absolute measure of generalization across all possible datasets in Section 2.1, as follows:
>
> *”The AU-GOOD metric allows for the evaluation of model generalisation relative to a target deployment distribution. Therefore, it allows us to determine that, though apparently Model B has a more consistent performance through different similarity thresholds, Model A is a better choice for Target 2 which is skewed towards molecules similar to the training data. Conversely, Model B is a better choice for Target 1.”*
>
> **Q8. Are there any plans to extend the work to larger datasets and deep learning models? [...].**
>
> **A8.** Yes, we are working on benchmarking state-of-the-art models, particularly for protein representation learning, which have dataset sizes much larger than the ones considered here and where modern architectures would presumably perform better.
>
> **Q9. Table 1, I think the ranking should be 2,4,1,4 for the models read left to right?**
>
> **A9.** We apologize for the inconvenience, but we do not understand the question. We would appreciate it if you could clarify in more detail what you mean.
>
> - - -
> We thank you again for the thoroughness of your review as well as the insightful comments, we believe that they have led to a major improvement in the quality of the revised manuscript.

---

> > ### Author Response · Authors · 2024-11-26
> >
> > Thank you for your constructive feedback on our submission. We have revised the manuscript and provided a detailed response to address your concerns. We kindly ask you to review the updates and let us know if they resolve your comments. Your insights have been invaluable, and we truly appreciate your time and effort. Please feel free to share any further suggestions. Thank you again for your support!

---

> ### Comment · Reviewer_53DR · 2024-11-26
>
> Thank you for your reply. And thank you for the additional work put in.
>
> The authors have addressed the issues I have brought up. I will update my score to 6.

---

### Official Review · Reviewer_b5Gp · 2024-11-04

**Soundness:** 4
**Presentation:** 3
**Contribution:** 3
**Rating:** 8
**Confidence:** 3

**Summary:**

This paper proposed an interesting framework for evaluating the model generalization to the out-of-distribution data (OOD) specifically with-in the biochemical domain. The author first proposed a metric that measures the model generalization based on existing similarity measures. A dataset partitioning algorithm and non-parametric statistical testing methods are proposed to complete the framework for model generalization comparisons.

**Strengths:**

1. This paper presents a comprehensive framework that not only measures generalization capabilities but also provides a statistical method for comparing different models based on the new AU-GOOD metric​

2. In the experimental section and the Appendix, the authors present a detailed and systematic analysis of selecting the similarity function, including its crucial relationship with the robustness of the GOOD metric in predicting the OOD data.

3. The paper includes extensive experiments on multiple use cases to demonstrate the framework's effectiveness. Including selecting the model with more generalization abilities in case 1: predicting pharmaceutical properties of small molecules and case 2: predicting protein sequence properties.

4. The open source code repo is clean and well-organized.

**Weaknesses:**

1. I am a bit confused by the formulation process in section 2.1 and found the formulation process is somewhat misleading. Here is a simple example, given four data points with features x1, x2, x3, x4. Assume the similarity score is: s(x1, x2) = 0.2. s(x1, x3) = 0.3, s(x1, x4) = 0.5, s(x2, x3) = 0.7, s(x2, x4) = 0.8, s(x3, x4) = 0.9. Assume the maximal value of similarity \lambda between the training and testing elements is 0.85. In this case, \lambda_{s}=0.85 only restricts x3 and x4 to be both in either the training set or the test set. Therefore, \lambda_{s}=0.85 results in different partitions of these four data points. E.g (x1, x3, x4) for training (x2) for testing or (x3, x4) for training and (x1, x2) for testing, etc. I am wondering if it is proper to formulate eq.3 from eq.2 by eliminating the \phi, since \lambda_{s} can only  be considered as a partition constraint rather than partition method.

2. In addition, assuming \phi is not omitted in equation 5, and empirical risk R is rewrite as (say) a function of \phi. This indicates that, under the same threshold, the AU_GOOD value is not only dependent on the target distribution, but also the partition algorithms. It’s better to discuss different partition algorithms, and how it impacts the AU_GOOD, or, justify that the proposed CCPart can successfully partition the Out-of-distribution sets.

---

Minor concerns

3. For equation 2, \mathcal{T} denotes the training subset, and the empirical risk is calculated among the test set \mathcal{E}, the subscript of x should not be i.

4. It’s better to enlarge the captions, x-labs and y-labs in Figure 1.

5. Another minor shortcoming of this paper is that it lacks a ‘ground truth label’ for model generalization comparison. In use case 1, Figure 5 shows that the Random Forest is significantly better than the rest models in OOD generalization. However, this conclusion cannot be further verified. It is better to know exactly which model does better in handling the actual OOD data. (Though it is well recognized that RF and lightgbm is good at handling the outliers)

**Questions:**

My questions are listed in the weakness part.

---

> ### Author Response · Authors · 2024-11-21
> **Response to Reviewer b5Gp**
>
> We thank reviewer b5Gp for their insightful comments, as well as the recognition of our work. We address your concerns in the following points:
>
> ---
>
> **Q1. [...] I am wondering if it is proper to formulate eq.3 from eq.2 by eliminating the \phi, since \lambda_{s} can only be considered as a partition constraint rather than partition method.**
>
> **A1**. We agree with the reviewer in their analysis: unbiased partitioning of the dataset, given a fixed similarity threshold, is independent of the algorithm itself, and thus equations 2 and 3 should only depend on the similarity threshold. However, we think it is important to note that in practice, sampling the data so that it satisfies the partition constraint requires the use of a particular partitioning algorithm, which may not perform an unbiased sampling. Accordingly, we have changed the corresponding equations, and included the following comment at the beginning of Section 2.2:
>
> *”The AU-GOOD values can also be affected by the sampling performed during the partitioning step, as it may bias the composition of the test set.”*
>
> **Q2. [...]. It’s better to discuss different partition algorithms, and how it impacts the AU_GOOD, or, justify that the proposed CCPart can successfully partition the Out-of-distribution sets.**
>
> **A2.** We acknowledge the importance of the evaluation of the partitioning algorithm and we have included further clarification behind the CCPart algorithm in Section 2.2, as follows:
>
> *”There are three conditions that the partitioning algorithms have to fullfil: 1) to not remove any data point, otherwise the comparison between thresholds could also correspond to the loss of information and would make them not directly comparable; 2) to enforce strict boundaries between training and testing subsets for any given similarity threshold (Walsh et al., 2020), otherwise we would not be measuring strictly the dependence on the similarity threshold, but also (implicitly) the effect of overlap within any given threshold; and 3) to allow for multiple thresholds, otherwise we cannot describe model performance as a function of the threshold.*
>
> *All prior partitioning algorithms violate at least one of the three conditions: GraphPart (Teufel et al., 2023) or Lo-Hi splitting (Steshin, 2023) do not fulfill condition 1; perimeter split, maximum dissimilarity split (Ramsundar et al., 2019), the Butina split (Butina, 1999), Dyonisus (Tom et al., 2023), and Spectra (Ektefaie et al., 2024) do not fulfill condition 2; finally, scaffold split does not fulffil condition 3.”*
>
> Additionally, we have included a comment acknowledging the limitations of our study, but highlighting the versatility of our framework with respect to the partitioning algorithm, also in Section 2.2, as follows:
>
> *”In any case, our framework is agnostic to the specific partitioning strategy used and future development of alternative algorithms that fulfill the three conditions could improve the accuracy of the AU-GOOD calculation. It is also an open research question what the impact is of violating any of the three assumptions on the reliability of the AU-GOOD metric.”*
>
> Secondly, regarding the demonstration that CCPart can successfully partition out-of-distribution sets, we provide a qualitative empirical proof in the form of UMAP visualizations  of the chemical space in Section 3.1.1 and Appendix - F. We have renamed the heading to clarify this point from *“Visualisation of the partitioning algorithm”* to *“Qualitative analysis of CCPart as an OOD partitioning algorithm”*.
>
>  **Q3. For equation 2, \mathcal{T} denotes the training subset, and the empirical risk is calculated among the test set \mathcal{E}, the subscript of x should not be i.**
>
> **A3.** We have fixed the issue in the revised manuscript.
>
> **Q4. It’s better to enlarge the captions, x-labs and y-labs in Figure 1.**
>
> **A4.** We apologise for the inconvenient formatting and thank you for bringing it to our attention. We have reformatted Figure 1 to make it more readable.
>
> **Q5. Another minor shortcoming of this paper is that it lacks a ‘ground truth label’ for model generalization comparison. [...].**
>
> **A5.** We agree with you on this issue being a limitation of our study. We have incorporated a comment regarding this limitation in Section 2.1, as follows:
>
> *“Regarding the empirical comparison between methods, neither Tossou et al. (2024) nor Ektefaie et al. (2024) have provided any empirical evaluation of the properties of their method, due to the lack of available ground truth data. It is an open research question how to properly measure the accuracy of the methods for OOD evaluation. Further, each of the three methods evaluates slightly different properties of the models and it is unclear how a direct comparison could be performed.”*
>
> - - -
> We thank the reviewer for highlighting these issues and we hope that the revisions we have introduced to the manuscript have sufficiently addressed them.

---

> > ### Author Response · Authors · 2024-11-26
> >
> > Thank you for your constructive feedback on our submission. We have revised the manuscript and provided a detailed response to address your concerns. We kindly ask you to review the updates and let us know if they resolve your comments. Your insights have been invaluable, and we truly appreciate your time and effort. Please feel free to share any further suggestions. Thank you again for your support!

---

> > > ### Comment · Reviewer_b5Gp · 2024-11-26
> > > **Reply to the Author**
> > >
> > > Thanks for providing more information about the partitioning algorithms and for your efforts to address all my concerns. I have increased my score from 6 to 8. Since I am not familiar with this paper's first use case, I will keep my confidence score unchanged.

---

### Author Response · Authors · 2024-11-21
**General response to reviewers' comments**

Dear Reviewers,

Thank you for your valuable feedback. We acknowledge the validity of your comments and suggestions and after careful consideration we have decided to introduce revisions to our manuscript.

To make it easier for you to identify the modifications, changes to the text will be highlighted in yellow. Specific changes in response to individual reviewers are addressed in the respective discussions. We now summarise three key changes:

- To address the concern about the lack of monotonicity in the GOOD curves of certain datasets we have introduced new experiments exploring alternative similarity metrics that do not rely on molecular fingerprints. This change affects Figures 3, 4 and S1-3, and Table 1.
- To address the concern that the GOOD curves cannot be assumed to be linear, and therefore the slope is not a good metric of the dependence between similarity thresholds and model performance, we have replaced it with a measure of monotonicity which does not assume linearity (Spearman’s $\rho$ correlation coefficient). This change affects Tables 1, S2, and S3; and Figures 3, S1, and S4.
- Mathematical formalisation of the AU-GOOD metric has been modified and moved to the appendix and it is substituted by a "plain English" explanation (Reviewer mZMB, and b5Gp).

Once again, we appreciate your time and effort in providing constructive feedback.

Kind regards,

---

### Meta-Review · Area_Chair_5qir · 2024-12-18

**Metareview:**

Among the reviewers, there was broad agreement that this work
is interesting and relevant, that is it applicable to a wide range of biological domains, and that the experimental evaluation is convincing.
There were some points of criticism raised, such as the choice of the performance metric, the missing inclusion of evolutionary information, or the sensitivity to noisy inputs. After the discussion phase, however, I had the impression that the strengths outweigh the weaknesses, therefore I recommend acceptance of this paper.

**Additional Comments On Reviewer Discussion:**

The more conceptual points of criticism, such as the  choice of the performance metric, could be addressed in a convincing way during the rebuttal and discussion phase.

---

### Decision · Program_Chairs · 2025-01-22

Accept (Poster)